# Tree hydrodynamic modelling of the soil plant atmosphere continuum using FETCH3

Marcela Silva[1], Ashley M. Matheny[2], Valentijn R.N. Pauwels[1], Dimetre Triadis[3,4], Justine E. Missik[5], Gil Bohrer[5], and Edoardo Daly[1]

[1]Department of Civil Engineering, Monash University, Clayton, VIC, Australia

[2]Department of Geological Sciences, Jackson School of Geosciences, University of Texas at Austin, TX, USA

[3]Department of Mathematics and Statistics, La Trobe University, Bundoora, VIC, Australia

[4]Institute of Mathematics for Industry, Kyushu University, Fukuoka, Japan

[5]Department of Civil, Environmental and Geodetic Engineering, Ohio State University, OH, USA

**Correspondence:** Marcela Silva (marcela.defreitassilva@monash.edu), Edoardo Daly (edoardo.daly@monash.edu)

**Abstract.** Modelling the water transport along the soil-plant-atmosphere continuum is fundamental to estimating and predicting transpiration fluxes. A tree-hydrodynamic model (FETCH3) for the water fluxes across the soil-plant-atmosphere continuum is presented here. The model combines the water transport pathways to one vertical dimension, and assumes that the water flow through the soil, roots, and above-ground xylem can be approximated as flow in porous media. This results in a system of three partial differential equations resembling the Richardson-Richards equation describing the transport of water through the plant system and with additional terms representing sinks and sources for the transfer of water from the soil to the roots and from the leaves to the atmosphere. The numerical scheme, developed in Python 3, was tested against exact analytical solutions for steady state and transient conditions using simplified but realistic model parametrizations. The model was also used to simulate a previously published case study where observed transpiration rates were available in order to evaluate model performance. With the same model setup as the published case study, FETCH3 results were in agreement with observations. Through a rigorous coupling of soil, roots xylem, and stem xylem, FETCH3 can account for variable water capacitance while conserving mass and the continuity of the water potential between these three layers. FETCH3 provides a ready-to-use open access numerical model for the simulation of water fluxes across the soil-plant-atmosphere continuum.

## 1 Introduction

Plant transpiration drives the exchange of water and energy between the land and atmosphere (Katul et al., 2012), influencing ecosystem carbon uptake, as well as the partitioning of rainfall into evapotranspiration, runoff, and groundwater recharge.

Transpiration fluxes are driven by complex biological and physical processes, which, by interacting with each other, link the soil to the atmosphere through the above- and below-ground structures of plants. Several approaches exist to model the transpiration fluxes from the soil to the atmosphere (Fatichi et al., 2016; Matheny et al., 2017; Mencuccini et al., 2019).

Most models that explicitly resolve the movement of water within the plant system rely on the cohesion-tension theory, which explains how water can be transferred upward from the soil to the atmosphere across a tree height of several meters, in the absence of osmotic pressure differences (Couvreur et al., 2018). An uninterrupted water column can extend from the roots to the leaves under tension and, as the stomata open, water is transferred to the atmosphere pulling water from the soil, through the roots and xylem (Steudle, 2001). Accordingly, the system composed by the Soil, Plant, and Atmosphere is interpreted as

a Continuum (SPAC) with water flowing through its different compartments following a path of decreasing water potentials (Nobel, 2009).

In this context, the first models proposed to describe transpiration fluxes used an electrical analogy, with water flowing from one compartment to the other following water potential gradients associated across plant conductive tissue with resistances to the flow (van den Honert, 1948; Cowan, 1965; Sperry et al., 2003; de Jong van Lier et al., 2008; Jones, 2009). Recent

advances in these models account for the water storage within the plant using capacitors, and link the water and $CO_2$ fluxes through the stomatal conductance (Cruiziat et al., 2002; Daly et al., 2004b, a; Manzoni et al., 2013; Bartlett et al., 2014; Manoli et al., 2014; Hartzell et al., 2018). Electric-circuit models commonly assume that the water flow along the SPAC occurs as a succession of steady states, whereby the water potentials in the different compartments of the system adjust instantaneously to environmental changes. Many electric-circuit models also treat the soil as a finite capacity and often consider a single

compartment for each plant component (e.g., root and stem xylem) (Daly et al., 2004a; Hartzell et al., 2017). A finer resolution of resistances and capacitances might be used if a more detailed representation is desirable, but adding more layers may yield ordinary differential equations that are more difficult to solve (Chuang et al., 2006; Fatichi et al., 2016). A few electric-circuit models include formulations that account for root water compensation and other traits, although such inclusion requires the introduction of empirical parameters in the root water uptake formulation (Couvreur et al., 2012; Meunier et al., 2018; Kennedy

et al., 2019).

A continuous representation of the SPAC can be achieved in models that describe the water flow in the soil and plant xylem as flow in porous media (Fruh and Kurth, 1999). Porous-media models combine the continuity equation with the Darcy's law to define partial differential equations for the unsteady dynamics of the water potential across the SPAC and account for the transient response of water potential along the tree system. Some applications of these models focus on the water fluxes within

45 the above-ground stem (Kumagai, 2001; Bohrer et al., 2005; Chuang et al., 2006), others are centred on the simulation of below-ground fluxes and the interaction between soil and roots (Somma et al., 1998; Mendel et al., 2002; Amenu and Kumar, 2007; Teodosio et al., 2017), with more recent applications looking at the whole SPAC system (Janott et al., 2011; Verma et al., 2014; Quijano and Kumar, 2015; Mirfenderesgi et al., 2016; Huang et al., 2017). Porous-media models are able to simulate a variety of processes, such as root water compensation and hydraulic redistribution (Verma et al., 2014), which are

50 embedded in the root water uptake formulation. A canopy representation can also be accomplished by accounting for a leaf

area distribution and light distribution functions throughout the stem (Christoffersen et al., 2016), and dynamic formulations for the stem capacitance and conductances can be considered (Mirfenderesgi et al., 2018).

Porous-media models that simulate the entire tree structure, with a detailed 3D representation of branches and root systems, are computationally demanding and require specific and complex parameterizations. As a result, application of these models is impracticable to simulate water flow in more than a single tree (Bohrer et al., 2005; Janott et al., 2011). One-dimensional models that lump within-tree spatial hydraulic variability in their parameters are a more practical option to represent water movement in individual trees and within stands (Amenu and Kumar, 2007; Quijano and Kumar, 2015; Mirfenderesgi et al., 2016, 2018).

Another axis of complexity that differentiate transpiration models is the level of vertical detail of the canopy representation. Single-leaf models represent the simplest approach and resolve evaporative demand from the canopy as a single surface. More advanced approaches represent the canopy as two layers, of light and shade leaves, or as multiple layers, each of a different type/size cohort of trees within the canopy (e.g., Medvigy et al., 2009). Advances of canopy representation include the development of vertically detailed canopy representations (e.g., Drewry et al., 2010; Chen et al., 2016; Bonan et al., 2018), which led to a strong call to advance global land surface models by including a multi-layered canopy representation (Bonan et al., 2021). The complexity of the vertical representation of the canopy for the purpose of light attenuation and atmospheric demand for water could be decoupled from the complexity of the vertical representation of the hydraulic conductive pathway. For example, some models include vertically detailed canopy but represent the hydraulic pathway at its most rudimentary simplistic form as a set of three (soil, xylem, and leaf) reservoirs (Trugman et al., 2016; Xu et al., 2016). While this approach is more numerically efficient, it may lose some of the stomata control dynamics that is expressed due to different rates of water storage losses at different elevations through the canopy. Specifically, it was shown that the higher leaves would experience water limitations due to storage loss sooner in the day than the lower leaves (Bohrer et al., 2005). Conversely, the continuous vertically detailed system of partial differential equations solved by porous-media models makes them a better choice to simulate plant hydraulic behavior, species-specific hydraulic traits, and their interactions with environmental drivers across different species and ecosystem types (Matheny et al., 2017), providing a more detailed representation of the tree domain and canopy structure effects than electric-circuit models or single-layer porous-media models.

The aim of this study is to present FETCH3, an open source and open access tree hydrodynamic model for the simulation of the temporal and vertical dynamics of water storage and fluxes from the soil to the atmosphere, accounting for the vegetation response to environmental conditions and soil water availability. As a porous-media model, FETCH3 solves a system of three partial differential equations in a 1D domain to describe the water flow through the soil, root xylem, and stem xylem. The primary novelty of the model is a full coupling of the soil, roots, and stem xylem by clarifying the links between these 3 components of the system when re-scaling the processes into a single, continuous vertical dimension. The numerical formulation of FETCH3 was verified against exact solutions of simplified expressions of the equations, the model performance was evaluated against observational data collected during six months from a case study, and the inclusion of details of the canopy structure and stem xylem capacitance is discussed.

## 2 Model description

### 2.1 Model overview

The Finite-difference Ecosystem-scale Tree Crown Hydrodynamics (FETCH3) builds upon FETCH2 (Mirfenderesgi et al., 2016, 2018), which is based on its precursor, the Finite Element Tree Crown Hydrodynamics (FETCH) model (Bohrer et al., 2005). FETCH simulates water flow along a tree's stem and branches accounting for the branch structure in three dimensions. Simulating the three-dimensional tree crown structure is computational demanding and can solely be applied to a single tree. As a result, FETCH2 was developed to offer a more mechanistic approach that could be scaled to entire ecosystems. To achieve this, FETCH2 simplifies branches along the vertical direction, leading to a 1D model; the equations in FETCH2 are solved using a finite difference scheme (Mirfenderesgi et al., 2016).

Similarly to FETCH and FETCH2, FETCH3 assumes that the water movement in the xylem resembles flow in porous media; as in FETCH2, a macroscopic approach is used to simulate the water fluxes across the soil, roots, and stems with the fluxes being described in one dimension along the vertical direction (Fig. 1). As a development from FETCH2, FETCH3 presents a clearer link between the three different components of the system (i.e., soil, roots and stem), based on the conservation of water in each of the components, as derived in the Supplementary Material. In its 1D domain, FETCH3 allows for the vertical variation of the soil, root xylem, and stem xylem hydraulic parameters, which are able to vary along the tree. As a result, when combined, the quantities in the equations for the roots and stem are scaled to a reference ground area, consistently with the Richardson-Richards equation for the soil. This guarantees the conservation of mass as water flows from one component to the other. The system of equations in FETCH3 is also solved differently from FETCH2. As described in detail in the Supplementary Material, the equations in FETCH3 are discretised using the method by Celia et al. (1990) generating a system of algebraic equations combined into a single matrix, that is solved at the same time to guarantee the conservation of mass across the whole system comprising soil, roots and stem.

In FETCH3, water in a variably saturated soil is exchanged between the soil and the root system. The water flow in the soil is modelled using the Richardson-Richards equation with a term simulating the exchange of water between the soil and the roots. This term is a function of the difference in water potential between the soil and root layers; it thus results in a water sink during the day, when the water potential in the roots is low due to water loss by transpiration, but may act as a source of water to the soil during some nights, depending on the water content in different soil layers. The boundary conditions at the top and bottom of the soil column can be expressed as a flux or a value of soil water potential (refer to the Supplementary Material, section S.2.2).

Water fluxes within roots are likewise modelled with a Richardson-Richards type equation with the same term (of the opposite sign) representing water exchange between roots and soil. Soil and roots are coupled through this term, such that a sink of water in the soil is a source of water in the roots, and vice versa. The transfer of water between the soil and the roots is modulated by a conductance, representing the radial resistance between the bulk soil, roots surface, and root xylem, and a stress function, accounting for the reduction of the root water uptake associated with different soil moisture conditions possibly leading to water and oxygen stress. The 3D root architecture is scaled along the vertical dimension using a vertical mass distribution of

the roots and an index that summarizes the extent of lateral root area per unit of ground area (Quijano and Kumar, 2015). Water fluxes through the soil are defined as the mass flow of water per unit of ground volume. Thus, when referring the water fluxes in the roots to the same water mass that was contributed by the soil, the water storage and water fluxes within the roots must be re-scaled to the ground volume and thus, when normalized by unit depth, to the ground area.

A similar approach is used to model the water flow in the above-ground xylem, which is also described with a Richardson-Richards type equation with a sink term associated with transpiration losses from the canopy to the air. This equation is commonly used to simulate water flow for a single tree (Chuang et al., 2006); however, in order to correctly couple the above-ground and the below-ground components of the system, both equations must refer the water flux to units of ground area. This ensures the water mass balance and the continuity of the fluxes from soil through the root system to the above-ground stem xylem and ultimately to the air. This conservation of flux throughout the system is important but not trivial, as the amount of roots that fits within a reference area of soil, for example, is different than the xylem area or leaf area which are located above the same area of soil. FETCH3 simulates variable plant water storage below- and above-ground by using a dynamic capacitance function. Accounting for whole-plant water storage enables different model applications in which plant storage plays an important role, such as water use efficiency and plant hydraulic stress during dry periods (Huang et al., 2017; Li et al., 2021).

The complete system of equations simulates the water fluxes assuming a spatial distribution of tree, and their associated roots, stem xylem, and leaves, with an average cross-sectional area per unit of ground area. In this manner, FETCH3 presents a novel up-scaling technique required to properly calculate tree transpiration from small and large areas, such as a forest stand or plantations, assuming that all trees within the simulated area are similar on their dimensions and conductive parameters.

## 2.2 Governing equations

A detailed derivation of the equations is provided in the supplementary material and only the final set of equations is reported here for brevity.

Water flow in a variably saturated soil is described using the Richardson-Richards equation with a sink term for root water uptake:

$$C_s \frac{\partial \Phi_s}{\partial t} = \frac{d\theta_s}{d\Phi_s} \frac{\partial \Phi_s}{\partial t} = \frac{\partial}{\partial z} \left[ K_s \left( \frac{\partial \Phi_s}{\partial z} + \rho g \right) \right] - S, \tag{1}$$

where $C_s$ (Pa$^{-1}$) is the soil water capacitance, $\Phi_s$ (Pa) is the soil water potential, $\theta_s$ (m$^3$ m$^{-3}$) is the soil volumetric water content, $K_s$ (m$^2$ s$^{-1}$ Pa$^{-1}$) is the effective soil hydraulic conductivity, $\rho$ (kg m$^{-3}$) is the water density, $g$ (m s$^{-2}$) is the gravitational acceleration, $S$ ($s^{-1}$) is the root water uptake, and $t$ (s) is time, and $z$ (m) is distance along vertical direction, assuming positive represents upward flux. The relationships between $K_s$, $\Phi_s$ and $\theta_s$ are modelled according to van Genuchten (1980).

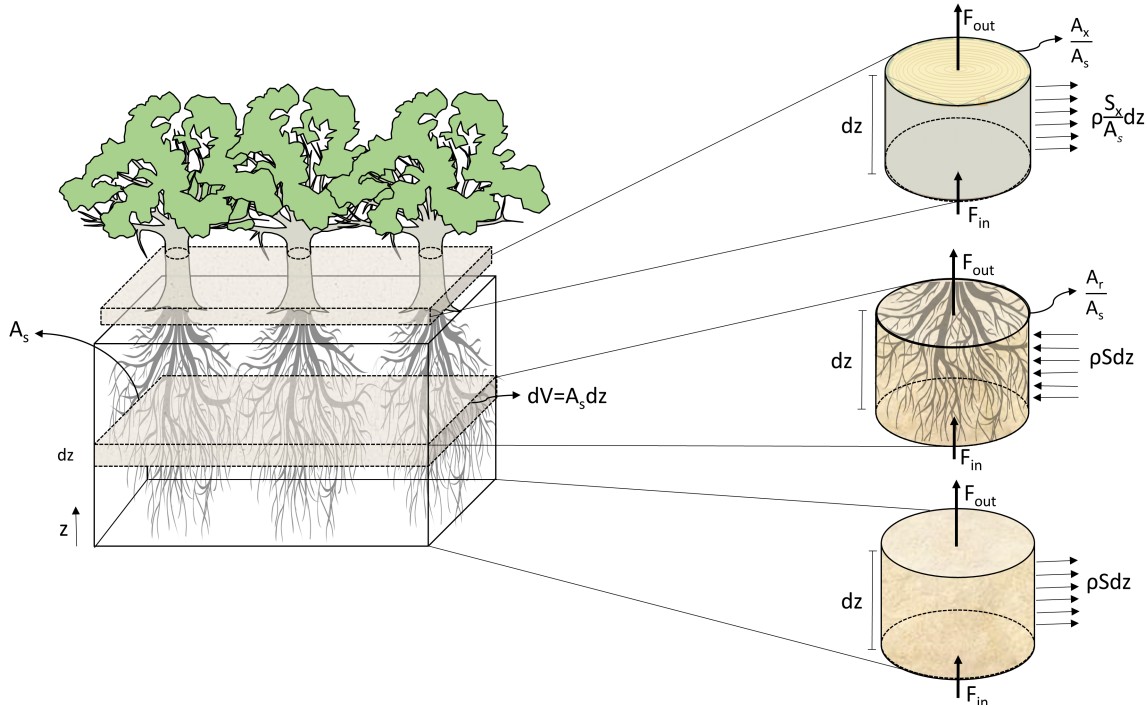

**Figure 1.** Representation of the coupling process between soil, root xylem, and stem xylem applied in the model, where $A_s$ represents a reference ground area, dz an infinitesimal depth over an area (m), z the vertical coordinate (m), V volume of soil (m$^3$), $\rho$ the density of water (kg m$^{-3}$), $F_{in}$ (kg s$^{-1}$) the water fluxes entering and $F_{out}$ (kg s$^{-1}$) exiting the volume, $A_r/A_s$ (m$^2_{root}$ m$^{-2}_{ground}$) the root xylem cross area index, $A_x/A_s$ (m$^2_{xylem}$m$^{-2}_{ground}$) the stem xylem cross area index, $S$ (s$^{-1}$) the rate at which water is extracted from the soil and enter the root xylem, and $S_x$ (m$^2$ s$^{-1}$) is the flow of water leaving the stem per unit of vertical length due to transpiration.

Considering the cross-sectional area of roots ($A_r$) per unit of ground area ($A_s$), the equation describing the water flow through
the roots reads

$$C_r \frac{\partial \Phi_r}{\partial t} = \frac{d}{d\Phi_r}\left(\frac{\theta_r A_r}{A_s}\right)\frac{\partial \Phi_r}{\partial t} = \frac{\partial}{\partial z}\left[K_r \frac{A_r}{A_s}\left(\frac{\partial \Phi_r}{\partial z} + \rho g\right)\right] + S, \tag{2}$$

where $C_r$ (Pa$^{-1}$) is the root xylem water capacitance, $\Phi_r$ (Pa) is the root water potential, $\theta_r$ is the root volumetric water content, $K_r$ (m$^2$ s$^{-1}$ Pa$^{-1}$) is the effective axial hydraulic conductivity of the roots, and $A_r/A_s$ (m$^2_{root}$ m$^{-2}_{ground}$) is the root cross sectional area index, representing the total root cross-sectional area at a given elevation per unit of ground area.
Above-ground, the flow through the cross-sectional area of stem xylem ($A_x$) per unit ground area is given by

$$C_x \frac{\partial \Phi_x}{\partial t} = \frac{d}{d\Phi_x}\left(\frac{\theta_x A_x}{A_s}\right)\frac{\partial \Phi_x}{\partial t} = \frac{\partial}{\partial z}\left[K_x \frac{A_x}{A_s}\left(\frac{\partial \Phi_x}{\partial z} + \rho g\right)\right] - \frac{S_x}{A_s}, \tag{3}$$

where $C_x$ (Pa$^{-1}$) is the stem xylem water capacitance, $\Phi_x$ (Pa) is the stem xylem water potential, $\theta_x$ (m$^3$ m$^{-3}$) is the stem xylem volumetric water content, $K_x$ (m$^2$ s$^{-1}$ Pa$^{-1}$) is the effective axial hydraulic conductivity of the stem xylem, $S_x$ (m$^2$ s$^{-1}$) is the flow of water leaving the stem per unit of vertical length due to transpiration, and $A_x/A_s$ (m$^2_{stem}$m$^{-2}_{ground}$) is the stem xylem cross-sectional area index. This index can be calculated from the tree sapwood area and stand density (typically reported for forest plots as number of trees per hectare), representing the total sapwood area per unit of ground area. The cross sectional area indicies applied to the roots and stem xylem guarantee the conservation of water as it flows across soil, roots, and stem.

### 2.2.1 Root water uptake and transpiration

Eqs. (1) and (2) are coupled through the exchange of water between the soil and roots. The term $S$ is modelled as a function of the difference between the water potential in the soil and the roots. This approach, introduced by Gardner (1960), was applied in several studies (Herkelrath et al., 1977; Mendel et al., 2002; Amenu and Kumar, 2007). Accordingly, $S$ (s$^{-1}$) is expressed as

$$S(z,t) = k_{s,rad}\, f(\theta_s(z,t)) \cdot \frac{A_{ls}}{A_s}(z) \cdot \frac{r(z)}{\int_{z_{r_i}}^{z_{r_j}} r(z)dz} \cdot (\Phi_s(z,t) - \Phi_r(z,t)), \tag{4}$$

where $k_{s,rad}$ (m$^3$s$^{-1}$m$^{-2}_{root}$Pa$^{-1}$) is the soil-to-root radial conductance per unit of root surface area, $f(\theta_s)$ is a dimensionless reduction function due to soil moisture, $r(z)$ the root mass distribution, with $z_{r_i}$ and $z_{r_j}$ (m) representing the elevation of the bottom and top of the roots, and $A_{ls}/A_s$ (m$^2_{root}$m$^{-2}_{ground}$) is an index defining the lateral root surface area per unit of ground, representing the root surface area taking up water from the soil. The vertical profile of root mass distribution represents the percentage of roots contained in different soil layer. The product of these two terms provide the portion of roots contributing to the exchange of water between soil and roots; this changes with depth depending on how the roots are vertically distributed. The water lost to the atmosphere is calculated using a transpiration function that depends on meteorological variables and limits the amount of water leaving the stomata as a function of the stem water potential. FETCH3 allows for the implementation of different transpiration functions, and a complete description of the transpiration formulation applied in this study is in the Supplementary Material, section S.3. Accordingly, $S_x/A_s$ (s$^{-1}$) reads

$$\frac{S_x}{A_s} = T \cdot l(z), \tag{5}$$

where $T$ (m s$^{-1}$) is the transpiration rate defined per unit of ground area, which is distributed along the canopy height via the leaf area density distribution ($l(z)$, m$^2$ m$^{-2}$ m$^{-1}$), which is the leaf area per unit of ground area per unit of height, which integrates vertically to the leaf area index (LAI). This effectively assumes that transpiration is proportional to leaf area throughout the depth of the canopy. We have found that in canopies where most leaves are concentrated near the upper layers, the results are not very sensitive to this simplification. More complex representations of the vertical distribution of transpiration through the canopy depth have been developed. Such vertically detailed canopy transpiration models assume, for example, that transpiration is vertically distributed proportionally to vertical light extinction through the depth of the canopy (Shaw and Schumann, 1992; Bohrer et al., 2009), or that transpiration rate is vertically distributed as a function that combines light attenuation and the

vertical profiles of other physical radiative forcing, such as turbulence, wind speed, temperature, and humidity (Bonan et al., 2018; Drewry et al., 2010). Such transpiration models can be easily implemented in FETCH3 by replacing Eq. (5) with a more elaborate vertical redistribution scheme, provided that the vertical descriptions of the required parameters for leaf area density, light attenuation and other physical forcing are available for the simulated forest plot.

## 2.3 Numerical scheme

Eqs. (1) - (3) are solved simultaneously using a finite difference numerical scheme, following Celia et al. (1990). The equations are solved using a fully implicit Picard method, with a backward Euler temporal discretization, as detailed in the Supplementary Material. The scheme is implemented in Python 3 in modular manner (i.e., with subroutines to define effective conductances and the water capacitance of the soil, roots, and stem xylem).

## 3 Model experiments

Three applications were used to i) test the correctness of the numerical scheme against analytical solutions, ii) compare results to a published case study, and iii) show the implementation of a leaf area density profile and a xylem capacitance function dependent on the xylem water potential.

### 3.1 Testing against analytical solutions

The numerical scheme was tested against three simplified cases that permit the derivation of solutions in closed form. Because of the nonlinear nature of the Richardson-Richards equation, only a few exact solutions are available, particularly when including sink and source terms (Broadbridge et al., 2017). An exact solution of the combined soil-to-air system (Eqs. 1 - 3) is thus too challenging to be derived. Therefore, the numerical scheme was tested against one of the equations. Eq. (3) was selected for this exercise and it was re-written as

$$\frac{\partial}{\partial t}\left(A_x\,\theta_x\right) = \frac{\partial}{\partial z}\left(A_x\,K_x\frac{\partial \Phi_x}{\partial z}\right) + \frac{\partial}{\partial z}\left(A_x\,K_x\rho g\right) - S_x, \tag{6}$$

where $S_x = l_a T$, with $T = T(\Phi_x, z, t)$ (m s$^{-1}$) the transpiration rate, and $l_a = l A_s$ (m) the leaf area per unit of height; $z$ (m) is bound between 0 at the bottom of the tree and $L$ at the top of the tree.

For analytical tractability of Eq. (6), simplified formulations of the hydraulic conductivity and water capacitance are used. The hydraulic conductivity is assumed to decrease with the water potential following the vulnerability curve (Bohrer et al., 2005; Chuang et al., 2006)

$$K_x = K_m\,e^{\alpha_0 \Phi_x}, \tag{7}$$

with $\alpha_0$ (Pa$^{-1}$) an empirical constant and $K_m$ (m$^2$ s$^{-1}$ Pa$^{-1}$) the maximum hydraulic conductivity. Eq. (7) implies that $d\Phi_x = dK_x/(\alpha_0 K_x)$. The xylem water content is assumed to depend on $\Phi$ according to

$$\theta_x = \theta_{res,x} + (\theta_{sat,x} - \theta_{res,x})\,e^{\alpha_0 \Phi_x}, \tag{8}$$

with $\theta_{res,x}$ (-) and $\theta_{sat,x}$ (-) being the residual and maximum water content of the stem xylem.

With the further assumption that $A_x = A_0 \exp(-\beta z)$, with $\beta$ (m$^{-1}$) an empirical allometric parameter, Eq. (6) can be re-written as

$$\gamma_0 \frac{\partial K_x}{\partial t} = \frac{1}{\alpha_0} \frac{\partial^2 K_x}{\partial z^2} + \left( \rho g - \frac{\beta}{\alpha_0} \right) \frac{\partial K_x}{\partial z} - \rho g \beta K_x - l_a T e^{\beta z}, \tag{9}$$

where $\gamma_0 = (\theta_{sat,x} - \theta_{res,x})/K_m$, (s Pa m$^{-2}$).

Assuming that at time $t = 0$ the water potential is $\Phi_x(z,0)$, the initial condition for Eq. (9) reads

$$K_x(z,0) = K_m e^{\alpha_0 \Phi_x(z,0)}. \tag{10}$$

The boundary condition at the bottom of the tree is defined by a time series of water potentials (i.e., $\Phi_0 = \Phi_x(0,t)$), which results in

$$K_x(0,t) = K_m e^{\alpha_0 \Phi_0(t)}. \tag{11}$$

The flux of water at the top of the tree ($z = L$) is zero, leading to the boundary condition

$$\left( \frac{1}{\alpha_0} \frac{\partial K_x}{\partial z} + K_x \rho g \right)_{z=L} = 0. \tag{12}$$

Solutions of Eq. (9) with initial and boundary conditions in Eqs. (10) - (12) can be obtained for different expressions of $T(z,t)$ for some cases as presented in the following sections.

### 3.1.1 Simplified unsteady case

An exact solution of Eq. (9) can be obtained by assuming $\beta = 0$ and considering that the gradient of water potentials is the main contributor to the water fluxes (i.e., neglecting the term $\partial_z(A_x \, K_x \rho g)$ in Eq. 6). With these assumptions, Eq. (9) becomes a linear diffusion equation with a sink term that can be re-written in compact form as $f_x(z,t) = l_a T$ and the boundary condition at the top of the tree reading $(\partial_z K_x/\alpha_0)_{z=L} = 0$.

A general solution of this equation can be written as (Polyanin, 2001)

$$K_x(z,t) = \int_0^L K_x(\xi,t) G(z,\xi,t) d\xi + \frac{1}{\gamma_0 \alpha_0} \int_0^t K_x(0,\tau) \left[ \frac{\partial}{\partial \xi} G(x,\xi,t-\tau) \right]_{\xi=0} d\tau + \int_0^t \int_0^L f_x(\xi,\tau) G(z,\xi,t-\tau) d\xi d\tau, \tag{13}$$

where

$$G(x,\xi,t) = \frac{2}{L} \sum_{n=0}^{\infty} \sin \left[ \frac{\pi(2n+1)x}{2L} \right] \sin \left[ \frac{\pi(2n+1)\xi}{2L} \right] \exp \left[ -\frac{\pi^2(2n+1)^2 t}{4L^2 \gamma_0 \alpha_0} \right]. \tag{14}$$

For a case where transpiration depends only on time, the sink is expressed as

$$f_x = T_m(1 - \cos(2\pi t/24)), \tag{15}$$

where $T_m$ is the maximum transpiration rate, $t$ is considered to be in hours, and it is assumed that $l_a = 1$ (m$^{-1}$).

A fixed potential, equal to 0 MPa, was considered at the bottom of the stem and along the vertical direction as initial condition. This solution was tested for a 6 m high tree with the parameters listed in Table 1. Comparisons between the exact and numerical solutions using the sink term in Eq. (15) are shown in Fig (2). The errors associated with the numerical solution are small, reaching a maximum of approximately $0.25 \cdot 10^{-3}$ MPa at the top of the tree. The error followed the pattern of transpiration, reaching its peak during day time and corresponding to a maximum error of 0.09% of the exact solution. The mass balance error equalled 0.05% of the total water entering the tree during the simulated 2 days. Similarly, the lowest error could be observed at night, when transpiration approaches zero. The numerical solution presents errors that change periodically. After the influence of the initial condition disappears, the errors remain stable in time.

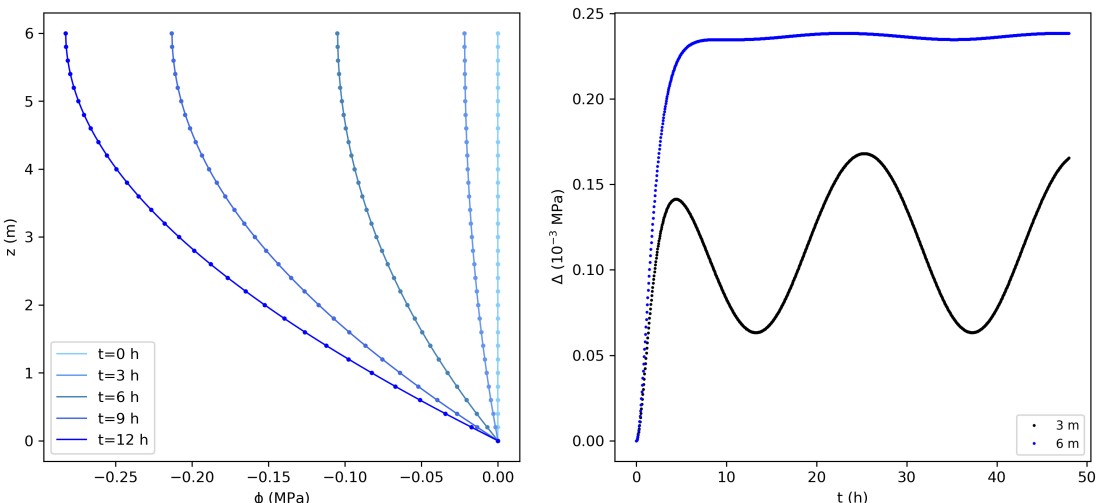

**Figure 2.** Left: water potentials (MPa) from the exact (lines) and numerical solutions (dots) using the sink term in Eq. (15) for the first 12 hours. For better visualization not all points are shown for the numerical solution. Right: difference between the exact and numerical solution ($\Delta$) at 3 m and 6 m. The temporal and spatial resolutions are 0.05 h and 0.01 m, respectively.

For a case where transpiration depends on both time ($t$) and the vertical position ($z$), the sink is written as

$$f_x = T_m \, z \, (1 - \cos(2\pi t/24)),\tag{16}$$

where $l_a = z$ (m$^{-1}$).

Comparisons between the analytical and numerical solutions using Eq. (16) are shown in Fig. 3, where 0 MPa was assumed at the bottom of the tree and as initial condition along the vertical direction. The error for this case is higher than for the previous case, with a maximum value that is about 0.2% of the exact solution, with a mass balance error equal to 0.05% of the total water entering the tree during the simulated 2 days. These errors would reduce using smaller values of $\Delta z$.

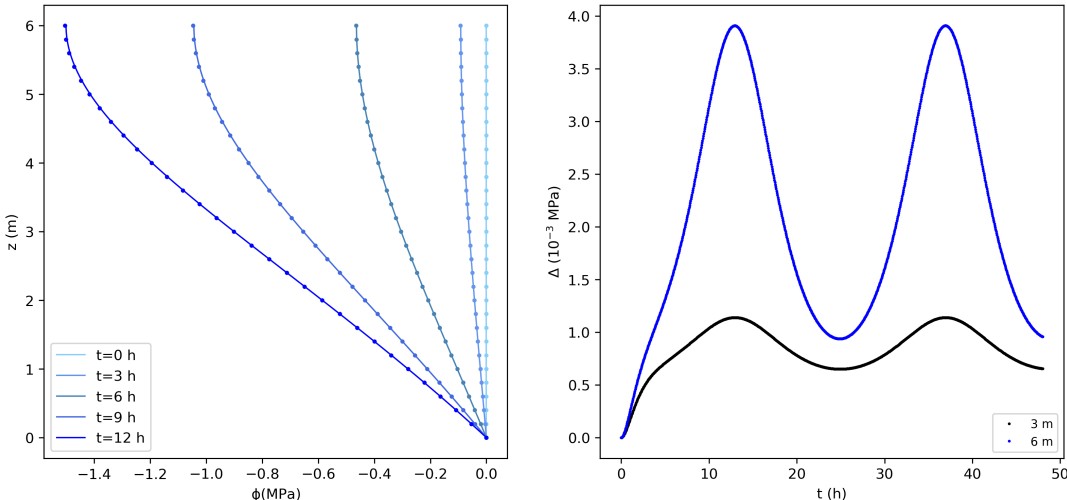

**Figure 3.** Left: water potentials (MPa) from the exact (lines) and numerical solutions (dots) using the sink term in Eq. (16) for the first 12 hours. For better visualization not all points are shown for the numerical solution. Right: difference between the exact and numerical solution ($\Delta$) at 3 m and 6 m. The temporal and spatial resolutions are 0.05 h and 0.01 m, respectively.

### 3.1.2 Steady-state solution

A solution of Eq. (9) at steady state can be obtained accounting for effects due to gravity and using a distribution of leaf area
260  per unit of stem height. It is assumed that the leaf area per unit of height is compatible with the Eq. (9) and satisfies $l_a(0) = 0$, a possible expression for $l_a(z)$ is

$$l_a(z) = \frac{l_m \beta_1}{\beta_2 - \beta_1} \left( \frac{\beta_2}{\beta_1} \right)^{\frac{\beta_2}{(\beta_2 - \beta_1)}} \left( e^{-\beta_1 z} - e^{-\beta_2 z} \right),\tag{17}$$

with $\beta_2 > \beta_1$. It is also assumed that the transpiration rate depends on the water potential and the elevation as

$$T = T_m e^{\alpha_0 \Phi_x} e^{\beta_1 - \beta}.\tag{18}$$

Accordingly, Eq. (9) at steady state reads

$$\frac{1}{\alpha_0} \frac{\partial^2 K_x}{\partial z^2} + \left( \rho g - \frac{\beta}{\alpha_0} \right) \frac{\partial K_x}{\partial z} - \rho g \beta K_x + \zeta (e^{-\eta - 1}) K_x = 0,\tag{19}$$

where $\eta = \beta_2 - \beta_1 > 0$ and

$$\zeta = \frac{l_m T_m \beta_1}{A_0 K_m (\beta_2 - \beta_1)} \left( \frac{\beta_2}{\beta_1} \right)^{\beta_2/(\beta_2 - \beta_1)}.\tag{20}$$

If it is assumed that the water potential initially has a generic profile and at the bottom of the tree remains constant in time, the
water potential will stabilize in time to a steady profile with the flux of water from the bottom of the tree equalling the flux of water being lost via transpiration.

The solution of Eq.(19) can thus be written as

$$K(z) = C_1 y^{(\alpha_0-\beta)/\eta} J_\upsilon(y) + C_2\, y^{(\alpha_0-\beta)/\eta} Y_\upsilon(y), \qquad (21)$$

where $J_\upsilon$ (.) and $Y_\upsilon$ (.) are the Bessel functions of the first and second kind (Abramowitz and Stegun, 1964) of order

$$\upsilon = \frac{[4\alpha_0\zeta + (\alpha_0 + \beta)^2]^{1/2}}{\eta}, \qquad (22)$$

and $C_1$ and $C_2$ are constants to be determined numerically by imposing the boundary conditions, as in Eqs. (11) and (12) with $\Phi_0$ constant.

The agreement between the exact and numerical solutions is shown in Fig. (4), for a case considering a bottom boundary condition of $\Phi_0$=0 MPa, a no-flux boundary condition at the top, and a hydrostatic initial condition. Steady state was reached after a short interval of about 3 hours of model time set. For a 6 meter high tree, the error of the numerical solution increases with elevation reaching approximately $0.4 \cdot 10^{-3}$ MPa at the tree top, being 0.4% of the exact value. According to the steady-state condition, the differences in storage between the last two consecutive model time steps approached zero and were equal to $-2.77 \cdot 10^{-18}$ m$^3$, with transpiration equalling 99.97% of the total flux entering the tree. A larger error was reached in comparison to the unsteady state solution cases due to the more complex formulation used for the steady-case scenario.

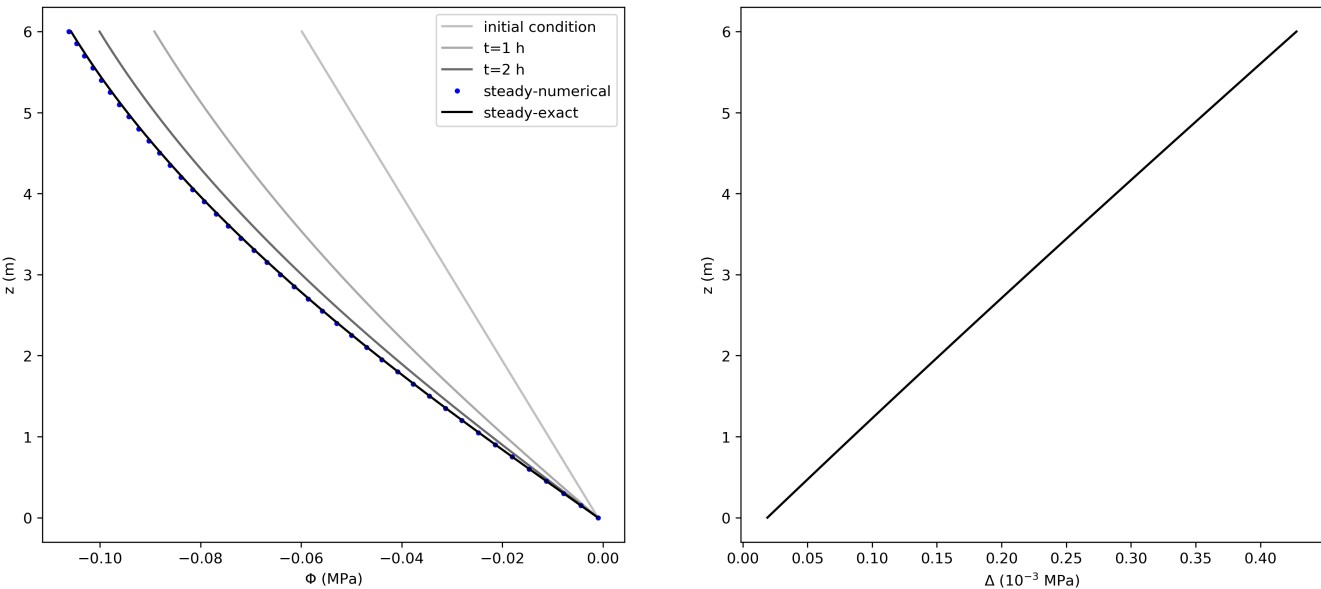

**Figure 4.** Left: water potentials, $\Phi$, (MPa) at steady state obtained from the exact (black line) (Eq. 21), and numerical solutions (dots), using 0.05 m and 0.08 h as spatial and temporal resolution, respectively. For the numerical solution, not all points are shown for better visualization. The lines with light colors present the initial condition and the first 2 hours of simulation. Right: difference between the exact and numerical solution ($\Delta$) at steady state condition along tree height.

**Table 1.** List of parameters used in the comparison between the exact and numerical solutions (section 3.1)

| Parameters | Value | Units | Description |
|---|---|---|---|
| $\beta$ | 0 | $\text{m}^{-1}$ | Allometric parameter, vertical reduction rate of cross-sectional area |
| $\beta_1$ | 0.2 | $\text{m}^{-1}$ | Empirical shape parameter for vertical leaf area distribution per unit xylem lenght |
| $\beta_2$ | 1 | $\text{m}^{-1}$ | Empirical shape parameter for vertical leaf area distribution per unit xylem lenght |
| $l_m$ | 0.2 | m | Mean leaf area per unit of length |
| $A_0$ | 0.0045 | $\text{m}^2$ | Basal cross-sectional area of the stem xylem |
| $T_m$ | $3.47 \cdot 10^{-8}$ | $\text{m s}^{-1}$ | Maximum transpiration rate |
| $\alpha_0$ | $5 \cdot 10^{-7}$ | $\text{Pa}^{-1}$ | Empirical constant |
| $\theta_{res,x}$ | 0.1 | - | Residual water content of the stem xylem |
| $\theta_{sat,x}$ | 0.6 | - | Saturated water content of the stem xylem |
| $K_m$ | $1.02 \cdot 10^{-9}$ | $\text{m}^2 \, \text{s}^{-1} \text{Pa}^{-1}$ | Maximum hydraulic conductivity |

## 3.2 Model application

FETCH3 was tested against a case study described in Verma et al. (2014). For reproducibility purposes, FETCH3 used the same model setup, environmental variables, and parameters as Verma et al. (2014), where the software COMSOL Multiphysics (Ver. 4.1; http://www.comsol.com/) was selected to solve the system of equations using finite elements. Details of the dataset are reported in Zeppel et al. (2008), Yunusa et al. (2012), and Verma et al. (2014), with a brief summary presented here.

### 3.2.1 Site description

The study site is located at latitude 33°39' 41" S and longitude 150°46' 57" E in New South Wales, Australia. According to the long term statistics (1993–2013 - Royal Australian Air Force base in Richmond, Australian Bureau of Meteorology, station 067105, http://www.bom.gov.au/climate/data/) the average daily minimum and maximum temperatures are 10 °C and 24 °C, with annual rainfall approximately 730 mm.

Rainfall, solar radiation, air temperature, and humidity were collected every 30 minutes from January 1st to June 4th in 2007. Sap flux data were collected for the same period, using the heat ratio technique at a half-hour resolution. The vegetation is dominated by *Eucalyptus parramattensis* C.A. Hall (Parramatta red gum) and *Angophora bakeri* E.C. Hall (narrow-leaved apple). The trees were 14 m tall on average, with a LAI between 1.3 and 1.9.

The soil is duplex, with a first layer up to a depth of 0.8 m being predominantly sand , with clay underneath. The soil parameters used in the model are listed in Table 2.

### 3.2.2 Model setup

The system of equations was solved for a soil depth of 5 m, and trees with a height of 14 m and root depth of 3.2 m. The boundary condition at the soil bottom was a constant water potential equal to -0.06 MPa, corresponding to a water content of

**Table 2.** List of soil parameters used in the model application

| Parameters | Units | Sand | Clay | Description |
|---|---|---|---|---|
| $k_{s,sat}$ | m s$^{-1}$ | $3.45 \cdot 10^{-5}$ | $1.94 \cdot 10^{-7}$ | Saturated hydraulic conductivity |
| $\theta_{sat}$ | – | 0.47 | 0.55 | Saturated volumetric soil moisture content |
| $\theta_{res}$ | – | 0.045 | 0.068 | Residual volumetric soil water content |
| $\alpha$[1] | m$^{-1}$ | 14.5 | 0.8 | van Genuchten parameter |
| $n$[1] | – | 2.4 | 1.5 | van Genuchten parameter |
| $\theta_1$[1] | – | 0.05 | 0.08 | Root water uptake reduction function parameter |
| $\theta_2$[1] | – | 0.09 | 0.12 | Root water uptake reduction function parameter |

[1] Definitions and equations using these parameters can be found in the Supplementary Material.

0.28 m$^3$m$^{-3}$. At the surface, measured rainfall was used as a flux boundary condition to compute soil water infiltration (refer
to the Supplementary Material, section S.2.2). The boundary conditions for the trees are a zero-flux condition at the bottom of
the roots and, above-ground, transpiration is applied as a boundary condition at the top of the canopy. Daytime transpiration is
modelled through the Penman Monteith equation (Allen et al., 1998) combined with a stomata conductance function (Jarvis,
1976), whereas night time transpiration follows a more simplified formulation composed of a constant nightime transpiration
value modulated by temperature, VPD, and water potential at night (see Supplementary Material for more details). In order
to follow the same setup as in Verma et al. (2014), transpiration is not distributed along the stem, but is imposed as a flux
concentrated at the top of the tree, and the water capacitance of the xylem in the roots and stem is assumed constant (Verma
et al., 2014).

In the sand layer, soil initial conditions are assumed to be a constant water potential equal to -0.004 MPa, corresponding to a
water content of 0.08 m$^3$m$^{-3}$. In the clay layer, water potential below a depth of 3 m was constant and equal to -0.06 MPa.
Between these two depths, water potential was interpolated linearly. For the tree, water potential linearly decreased from -0.06
MPa at the bottom of the roots to -0.22 MPa at the top of the canopy. The spatial resolution used was 0.1 m, and the time step
20 s. The list of parameters used in the model, including root water uptake and transpiration parameters, is in Table 3.

### 3.2.3   Results

The model predictions for sap-flux during the day compared well with observation during the entire measurement period
(Fig.5a), reaching a R$^2$ value of 0.74. The total mass balance error in the soil represented -0.30% of total infiltration, and it was
calculated as the change in soil water storage minus the difference between the flux entering (bottom boundary condition and
infiltration) and exiting the soil (root water uptake). In the tree (root and stem xylem), the water mass error was -0.16% of the
total infiltration, and was calculated as the change in water storage (in the stem and root xylem) minus the difference between
the fluxes entering (root water uptake) and exiting (transpiration) the tree. The model maintained a continuous water potential
along roots and stem xylem (Fig. 5b). At midday, in the roots, water potential decreases almost linearly with elevation, while
in the stem xylem, because of the transpiration flux at the top of the tree, it is non linear. The change in the gradient at the soil

**Table 3.** List of parameters used in the application of the model as in Verma et al. (2014)

| Parameters | Units | Value | Description |
|---|---|---|---|
| $A_x/A_s$ | - | $8.6 \cdot 10^{-4}$ | Stem xylem cross sectional area index surface ratio |
| $A_r/A_s$ | - | 1 | Root xylem cross sectional area index |
| $A_{ls}/A_s$ | - | 1 | Lateral root surface area index |
| LAI | - | 1.5 | Leaf area index |
| $k_{s,rad}$ | $\text{s}^{-1}$ | $7.2 \cdot 10^{-10}$ | Total soil-to-root radial conductance |
| $C_x$ | $\text{Pa}^{-1}$ | $1.1 \cdot 10^{-11}$ | Stem xylem water capacitance |
| $C_r$ | $\text{Pa}^{-1}$ | 1 | Root xylem water capacitance |
| $h$ | m | 14 | Tree height |
| $C_p$ [1] | $\text{J m}^{-3}\,\text{K}^{-1}$ | 1200 | Heat capacity of air |
| $T_{opt}$ [1] | K | 289.15 | Jarvis temperature parameter |
| $\lambda$ [1] | $\text{J m}^{-3}$ | $2.51 \cdot 10^9$ | Latent heat of vaporization |
| $\gamma$ [1] | $\text{Pa K}^{-1}$ | 66.7 | Psychometric constant |
| $g_b$ [1] | $\text{m s}^{-1}$ | $2 \cdot 10^{-2}$ | Leaf boundary layer conductance |
| $g_a$ [1] | $\text{m s}^{-1}$ | $2 \cdot 10^{-2}$ | Aerodynamic conductance |
| $k_r$ [1] | $\text{m}^2\,\text{W}^{-1}$ | $5 \cdot 10^{-3}$ | Jarvis radiation parameter |
| $k_t$ [1] | $\text{K}^{-2}$ | $1.6 \cdot 10^{-3}$ | Jarvis temperature parameter |
| $k_d$ [1] | $\text{Pa}^{-1}$ | $1.1 \cdot 10^{-3}$ | Jarvis vapor pressure deficit parameter |
| $h_{x50}$ [1] | m | -130 | Jarvis leaf water potential parameter |
| $g_{smax}$ [1] | $\text{m s}^{-1}$ | $10 \cdot 10^{-3}$ | Maximum leaf stomatal conductance |
| $n_l$ [1] | - | 2 | Jarvis leaf water potential parameter |
| $E_{max}$ [1] | $\text{m s}^{-1}$ | $1 \cdot 10^{-9}$ | Maximum night time transpiration |
| $q_z$ [1] | - | 9 | Root distribution parameter |
| $k_{max}$ [1] | $\text{m s}^{-1}$ | $1 \cdot 10^{-5}$ | Maximum conductivity of saturated stem xylem |
| $k_{sax}$ [1] | $\text{m s}^{-1}$ | $1 \cdot 10^{-5}$ | Specific axial conductivity for the root system |
| $a_p$ [1] | $\text{Pa}^{-1}$ | $2 \cdot 10^{-6}$ | Xylem cavitation parameter |
| $b_p$ [1] | Pa | $-1.5 \cdot 10^{-6}$ | Xylem cavitation parameter |

[1] Definitions and equations using these parameters can be found in the Supplementary Material.

surface is due to the sharp change in the axial hydraulic conductivity, since the xylem cross-sectional area index for the stem ($A_x/A_s=8.6\cdot 10^{-4}$) is different from that of the roots ($A_r/A_s=1$).

For the days shown in Fig.5b, when transpiration is peaking, the water potential fluctuates between a minimum of -2.2 MPa at the tree top and -0.8 MPa at the bottom of the roots. This range of values is in agreement with the results from the original studies and the published literature (Franks et al., 2007; Choat et al., 2012; Verma et al., 2014; Quijano and Kumar, 2015).

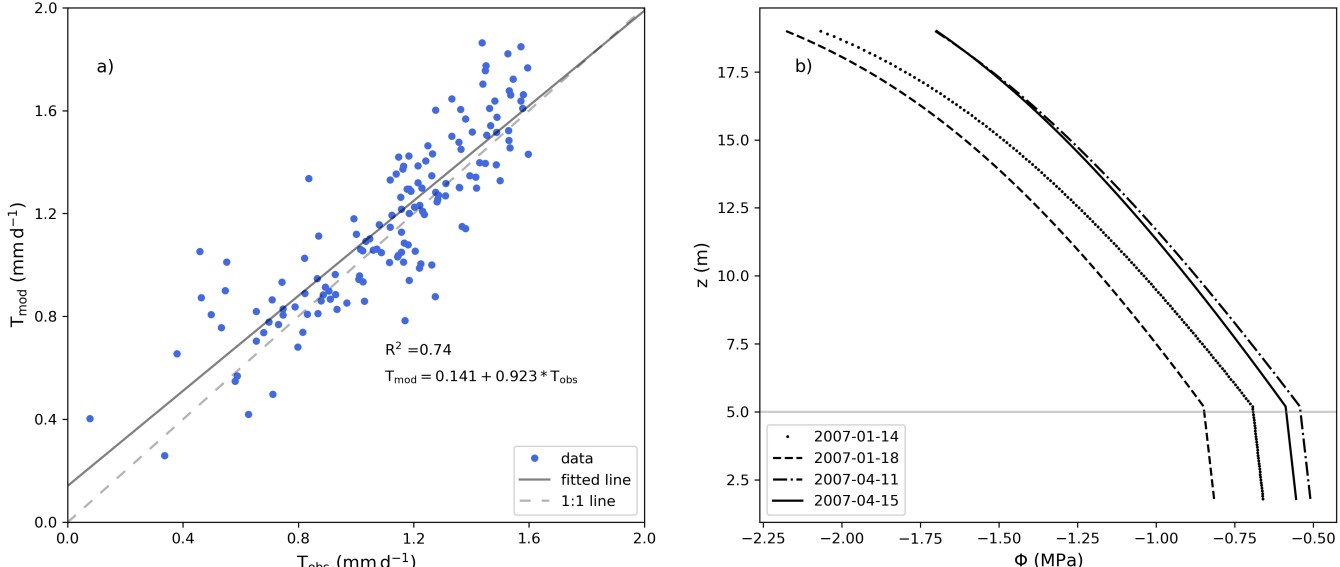

**Figure 5.** a) Comparison between measured ($T_{obs}$) and modelled ($T_{mod}$) daily sap flux rates excluding fluxes during the night. b) Root and stem xylem water potential (MPa) as a function of elevation (z) at midday. The vertical position of 5 m (above z=0, which is defined as the bottom of the soil column) represents the interface between between the roots and the stem.

A comparison of modelled and observed time series of transpiration rates for a week in January (summer) and April (autumn) is shown in Fig.6. The model is able to reproduce the temporal patterns of transpiration during the day, and does not show large fluxes at night because of the simplified modelling of the stomatal conductance at night, as in Verma et al. (2014) (see Supplementary Material).

FETCH3 was able to accurately represent the nonlinear interactions between the above- and below-ground components of the SPAC. From Fig 6, we can verify that root water uptake and transpiration are coupled, meaning that below-above ground interface is correctly represented by the model. Below-ground, shallow soil layers generated maximum rates of root water uptake (RWU) during most days, caused by greater root density and low water stress when water is readily available. During dryer days, with the decrease of soil moisture at the surface, considerable RWU was found in the deeper layers (approximately 20 - 30 cm from the soil surface). Root water uptake from deeper layers can be characterized as a hydraulic compensation path generated by rapid reductions in the top layers radial hydraulic conductivity, as it can be seen in Fig 6a, during the last 3 days in January.

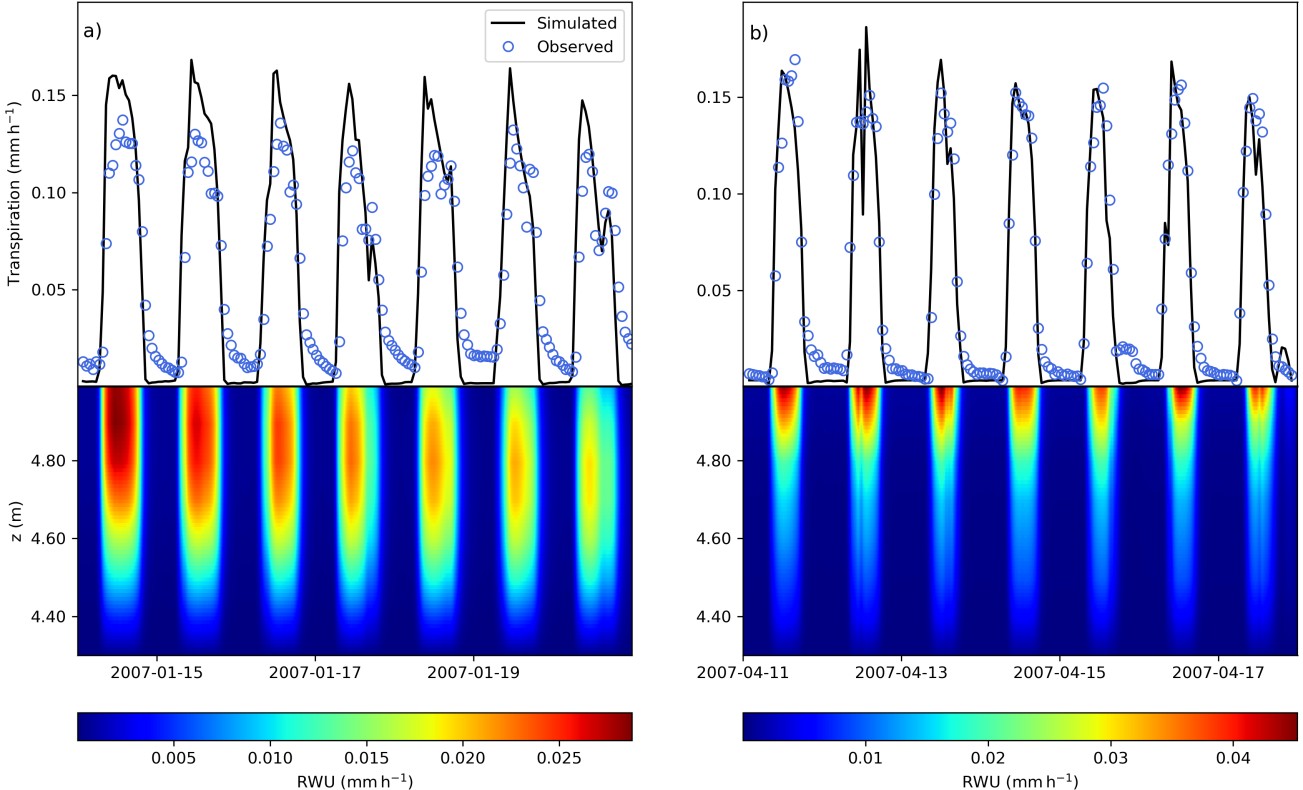

**Figure 6.** Comparison between modelled (black line) and observed (blue circles) transpiration rates and modelled root water uptake (colormap, $\mathrm{mm\,h^{-1}}$) during one-week periods in (a) January and in (b) April. The vertical position of 5 m (above z=0, which is defined as the bottom of the soil column) represents the interface between the roots and the above-ground stem xylem.

### 3.3 Modelling LAD and water capacitance

FETCH3 is able to simulate the distribution of transpiration along the vertical axis, as well as a water capacitance function for the roots and stem xylem. In order to test this capability, we applied FETCH3 using the same parameters and setup as in section 3.2.2, but changed how the transpiration and xylem water capacitance are modelled in the case study. For this experiment, transpiration is not a boundary condition at the tree top, but is distributed along the stem as in Eq. (3), with $S_x/A_s$ depending on the leaf area density (LAD). At the tree top, a no-flux condition is applied. An empirical LAD function described in Lalic and Mihailovic (2004) was used, and a LAD profile suitable for *Eucalyptus* stands can be written as:

$$l(z) = l_{max} \left( \frac{h - z_m}{h - z} \right)^{n_0} \exp \left[ n_0 \left( 1 - \frac{h - z_m}{h - z} \right) \right], \tag{23}$$

where $h$ is the tree height (m), $l_{max}$ ($\mathrm{m^2\,m^{-3}}$) is the maximum value of leaf area density in a layer, $z_m$ (m) is the corresponding above-ground height of $l_{max}$, and $n_0$ (-) is an empirical parameter defined as

**Table 4.** List of parameters used in the application of the model considering a water capacitance and a leaf area density function

| Parameters | Units | Value | Description |
|---|---|---|---|
| $l_{max}$ | $\mathrm{m^2 m^{-3}}$ | 0.4 | Maximum value of leaf area density |
| $z_m$ | m | 11 | Corresponding above-ground height of $l_{max}$ |
| $n_0$ | - | 6 or 0.5 | Empirical parameter (Eq. 24) |
| $\Phi_d$ | Pa | $5.74 \cdot 10^8$ | Empirical parameter for water pressure of dry xylem |
| $p$ | - | 20 | Empirical coefficient |
| $\theta_{sat,x}$ | - | 0.58 | Water content at saturation for the stem xylem |
| $\theta_{sat,r}$ | - | 0.58 | Water content at saturation for the root xylem |

$$n_0 = \begin{cases} 6 & 0 \leq z < z_m, \\ 0.5 & z_m \leq z \leq h. \end{cases} \quad (24)$$

The value of $l_{max}$ can be calculated from the LAI imposing

$$LAI = \int_0^h l(z)dz. \quad (25)$$

Following Chuang et al. (2006) and Bohrer et al. (2005), the water capacitance of the roots and stem xylem are

$$C_x(\Phi_x) = \frac{A_x}{A_s}\frac{\partial \theta_x}{\partial \Phi_x} = \frac{A_x p \theta_{sat,x}}{A_s \Phi_d}\left(\frac{\Phi_d - \Phi_x}{\Phi_d}\right)^{-(p+1)}, \quad (26)$$

$$C_r(\Phi_r) = \frac{A_r}{A_s}\frac{\partial \theta_r}{\partial \Phi_r} = \frac{A_r p \theta_{sat,r}}{A_s \Phi_d}\left(\frac{\Phi_d - \Phi_r}{\Phi_d}\right)^{-(p+1)}, \quad (27)$$

where $\Phi_d$ (Pa) and $p$ (-) are empirical coefficients for the hydraulic system, and $\theta_{sat,x}$ (-) and $\theta_{sat,r}$ (-) are the water content at
360 saturation for the stem and roots xylem, respectively. The values of these parameters are shown in Table 4.

From Fig. 7, the vertical distribution of transpiration follows the shape of the LAD, with larger values of transpiration where the
LAD is also large. Accordingly, $\Phi$ decreases along the tree height, in accordance with the no-flux boundary condition applied
at the top of the tree.

## 4   Conclusions

The Finite-difference Ecosystem-scale Tree Crown Hydrodynamics version 3 (FETCH3) was introduced in this study. By
using a porous-media approach, FETCH3 is able to simulate intradaily dynamics of transpiration and provides a fast response

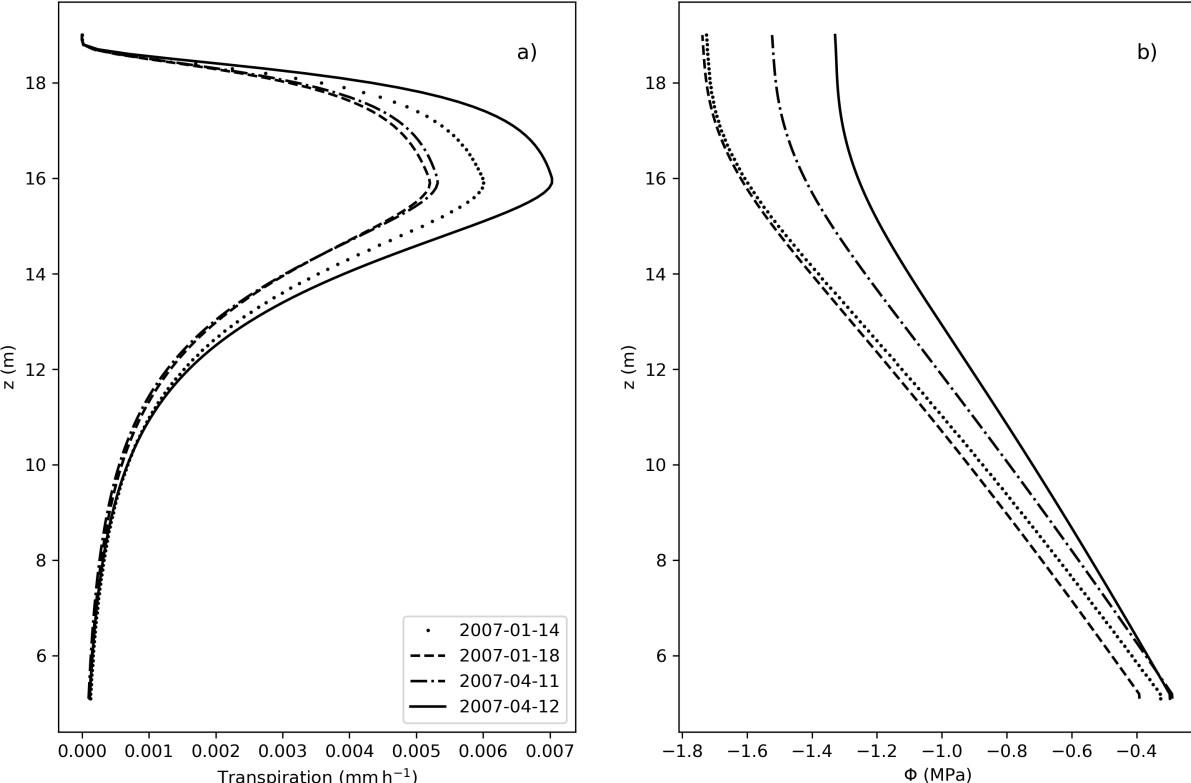

**Figure 7.** a) Transpiration fluxes $(\mathrm{mm\,h^{-1}})$ as a function of the elevation (z). b) Water potential (MPa) along z, considering z=0 at the bottom of the soil, and z=5 m equal the bottom of the stem.

to environmental variables. FETCH3 allows fidelity in the representation of hydraulic traits, which can be used to explore plant responses to water stress and xylem processes rather than land-atmosphere interactions.

We tested FETCH3 against exact numerical solutions of the equations and observations of transpiration. The numerical scheme

of the model was applied to two simplified exact non-steady state cases, reaching a maximum error of approximately 0.2% with respect to the exact solution at the tree top of a 6 m high tree, for a case in which transpiration is dependent on both time and elevation. For a steady-state scenario, considering a more complex formulation, the error approached 0.4% of the exact solution at the tree top.

Simulated transpiration rates from FETCH3 reached an $R^2$ of 0.74 in comparison to observed sapflow rates from a published

case study. In addition, values of water potential were continuous along roots and stem xylem, showing that water flux in the soil, roots, and stem are correctly coupled along the entire tree structure. By using a hydrodynamic set of equations, FETCH3 resolves the temporal and vertical dynamics of root water uptake, and stem and root water transport and storage. This allows FETCH3 to simulate hydrodynamic phenomena such as root water compensation following reductions of soil moisture in the shallow soil layers.

By comparing the model predictions of transpiration and soil and xylem water storage, with different sets of parameters (describing the whole-tree hydraulic strategy of the trees), and different environmental forcing (describing realistic or hypothetical conditions and stress), FETCH3 will allow model-based studies of the consequences of hydraulic traits and strategies of different tree species for above- and below-ground water transport, with a range of stem and root xylem hydraulic characteristics.

*Code and data availability.* The development of FETCH3 model and graphs presented in this paper were conducted in Python 3. The exact

version of FETCH3 used to produce the results can be found in the zenodo repository: https://doi.org/10.5281/zenodo.5775304. A more modular version of FETCH3, which also includes the formulation for a vertically distributed transpiration described in the model's previous versions, can be found in: https://doi.org/10.5281/zenodo.5775300.

*Author contributions.* MS, AMM, and ED designed the study; MS, VRNP, ED, and JEM developed the model scripts; AMM, GB, JEM, DT, VRNP, and ED supervised the writing and results; MS and ED wrote the original draft. All authors gave comments and contributed to the

390 final version of the manuscript.

*Competing interests.* The authors declare that they have no conflict of interest

*Acknowledgements.* ED was supported by the Australian Research Council through the Discovery Project DP180101229. GB and AMM were funded in part by NSF award 1521238. AMM was supported by the Department of Energy TES grant DE-SC0020116 and the National Science Foundation EAR CAREER award #2046768. GB and JEM were partially funded through BARD IS-5304-20.

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
