# Peer review of "Tree Hydrodynamic Modelling of Soil Plant Atmosphere Continuum (SPAC-3Hpy)"

_Geoscientific Model Development, 2021_

## Author Comment (AC3)

**Reviewer 2**

Reviewers' comments are in black font, our responses are in blue and green is for new and revised text; to assist with navigation, we use codes, such as R2C1 for Reviewer 2 Comment 1.

**General comments:**

In this manuscript, the authors describe the one-dimensional numerical model of tree hydrodynamics SPAC-3Hpy with explicit Richards-like solutions of flow in soil, root, and stem domains. It is proposed to validate it against analytical solutions of water flow in porous media, and against tree transpiration data. The scope is of great interest to the readership of Geoscientific Model Development. Overall, the study is interesting, well explained and provides clear illustrations.

We thank Dr Couvreur for his overall positive evaluation of our work.

My main concerns regarding publication in GMD are (i) the need for clarifications and small improvements that would facilitate the interpretation of the manuscript, (ii) the validation against transpiration data that I think does not meet the quality of the rest of the manuscript, and (iii) the lack of a proper contextualization of the model relative to the galaxy of existing tree hydrodynamics models that could be developed in a discussion section.

We have addressed the 3 points raised by the Reviewer in the answers to the specific comments below. In relation to the discussion section, we decided to add a section between the Introduction and the Model description to provide the context requested by both Reviewers.

To provide a clearer contextualization of the model, we decided to change the name of the model to FETCH3, and this is used in this document. Although the acronym no longer reflects the nature of the model and numerical scheme used to solve the equations, it easily relates to the developments that FETCH has had over the years.

**Specific comments:**

R2C1 (L11): The term "hydroactive" xylem is a bit odd as the process of water flow in xylem is passive as described in the introduction of the paper. Also, throughout the paper the authors seem to use the term "xylem" when referring to the stem (e.g. also at L84). This is confusing as there is xylem in roots and leaves too. I would suggest to revise the wording.

Thank you for the suggestion. We removed the term "hydroactive" xylem and introduced the terms "stem xylem" and "root xylem" to improve the reading.

R2C2, L12: I am enthused that the authors made their model open access. It also seems that the code is open source, which is even better I believe. If that is correct, why not making it explicit in the text?

Thank you for your comment. We added in the Introduction:

"The aim of this study is to present FETCH3, an open source and open access tree hydrodynamic model for the simulation of the temporal and vertical dynamics of water storage and fluxes from the soil to the atmosphere, accounting for the vegetation response to environmental conditions and soil water availability."

R2C3, L21-22: I think the sentence needs to be completed. Upward flow of water in 20 m trees would also be possible without the tension-cohesion mechanism, using an osmotic pressure difference of 0.2 MPa between soil and xylem. The tension-cohesion theory explains that this upward flow is possible without an osmotic driving force, along a continuum of liquid water under tension.

We added in the referred lines: "Most models that explicitly resolve the movement of water within the plant system rely on the cohesion-tension theory, which explains how water can be transferred upward from the soil to the atmosphere across a tree height of several meters, in the absence of osmotic pressure differences (**Couvreur et al., 2018**)."

R2C4, L40-42: Root water uptake compensation and hydraulic redistribution can also be modelled with "electric circuit" models that do not account for the plant water capacitance (see e.g. Meunier et al. (2017); Kennedy et al. (2019)). A major progress with the capacitance is that water fluxes along the stem do not have to be vertically equal simultaneously (e.g. as observed in Sperling et al. (2012) in palm) and that the integral of root water uptake does not have to match the integral of transpiration at a given time due to variations in stem water storage.

Thank you for your comment. We have added a new section between the Introduction and the Model description to provide a more detail review the existing literature on modelling water fluxes across the soil-plant-atmosphere continuum.

R2C5, L63-64: Similarities with FETCH and FETCH2 are mentioned but differences are unclear and should be clearly explained. A discussion section could as well discuss the specific interest of SPAC-H3py relative to the broader diversity of tree hydrodynamics models.

The differences between the present model, FETCH3, and FETCH2 are summarized in section 2.1 as:

"FETCH3 builds upon FETCH2 (Mirfenderesgi et al., 2016, 2018), which is based on its precursor, the finite element tree crown hydrodynamics (FETCH) model (Bohrer et al., 2005). FETCH simulates water flow along a tree's stem and branches accounting for the branch structure in three dimensions. Simulating the three-dimensional tree crown structure is computational demanding and can solely be applied to a single tree. As a result, FETCH2 was developed to offer a more mechanistic approach that could be scaled to entire ecosystems. To achieve this, FETCH2 simplifies branches along the vertical direction, leading to a 1D model; the equations in FETCH2 are solved using a finite difference scheme (Mirfenderesgi et al., 2016).

Similarly to FETCH and FETCH2, FETCH3 assumes that the water movement in the xylem resembles flow in porous media; as in FETCH2, a macroscopic approach is used to simulate the water fluxes across the soil, roots, and stems with the fluxes being described in one dimension along the vertical direction (Fig. 1). As a development from FETCH2, FETCH3 presents a clearer link between the three different components of the system (i.e., soil, roots and stem), based on the conservation of water in each of the components, as derived in the Supplementary Material. The links between the soil, roots, and stem xylem are clearer in FETCH3, thus providing a more precise coupling of the 3 components of the system. As a result, when combined, the quantities in the equations for the roots and stem are scaled to a reference ground area, consistently with the Richardson-Richards equation for the soil. This guarantees the conservation of mass as water flows from one component to the other. The system of equations in FETCH3 is also solved differently from FETCH2. As described in detail in the Supplementary Material, the equations in FETCH3 are discretised using the method by Celia et al. (1990) generating a system of algebraic equations combined into a single matrix, that is

solved at the same time to guarantee the conservation of mass across the whole system comprising soil, roots and stem."

Additionally, we have included in the manuscript a section to discuss the literature on plant hydraulic modelling, and the differences between the modelling approaches. This new section will clarify the pros and cons of each approach.

R2C6, Figure 1: It is good that variables appear in figure 1, so that connections to the equations can be done. However, several of these variables have not been defined at this point, which complicates the interpretation of the figure. There is room for it in the caption, or in the main text already.

The caption of Figure 1 has been modified as:

"Representation of the coupling process between soil, root xylem, and stem xylem applied in the model, where $A_s$ represents a reference ground area, dz an infinitesimal depth over an area (m), z the vertical coordinate (m), V volume of soil ($m^3$), ρ the density of water (kg $m^{-3}$), $F_{in}$ (kg $s^{-1}$) the water fluxes entering and $F_{out}$ (kg $s^{-1}$) exiting the volume, $A_r/A_s$ ($m^2_{root}$ $m^{-2}_{ground}$) the root xylem cross area index, $A_x/A_s$ ($m^2_{xylem}$ $m^{-2}_{ground}$) the stem xylem cross area index, S ($s^{-1}$) the rate at which water is extracted from the soil and enter the root xylem, and $S_x$ ($s^{-1}$) the rate of water leaving the stem xylem."

R2C7, L70-72: At this point it is not clear if radial resistances between the bulk soil, root surface and root xylem are considered. It is particularly important to specify it in the overview as they are frequently viewed as the largest resistances in series of the soil plant hydraulic continuum.

FETCH3 considers the radial resistances (which includes a soil moisture stress function), lateral surface area of the roots and root mass distribution when computing the root water uptake. We have added at line 71 of the overview section (2.1):

"Water fluxes within roots are likewise modelled with a Richardson-Richards type equation with the same term (of the opposite sign) representing water exchange between roots and soil. Soil and roots are coupled through this term, such that a sink of water in the soil is a source of water in the roots, and vice versa. The transfer of water between the soil and the roots is modulated by a conductance, representing the radial resistance between the bulk soil, roots surface, and root xylem, and a stress function, accounting for the reduction of the root water uptake associated with different soil moisture conditions possibly leading to water and oxygen stress."

R2C8, L99: I realize it is implicit, but probably worth specifying that the "soil capacitance" is the "soil water capacitance". The same goes for the root and stem water capacitances.

We have changed the terms throughout the manuscript as suggested.

R2C9, L99: Going through the equations I frequently wondered if parameters were constants or could vary in space. For instance, stem cross-sectional area, saturated hydraulic conductivity (here Km I believe) and sensitivity to cavitation (that I guess relate to the water capacitance parameter) vary along tree stems with substantial consequences on the nonlinear vertical water potential profile (Couvreur et al., 2018). It would be worth discussing briefly if such vertical variations of hydraulic parameters can be accounted for in SPAC-3Hpy.

We thank the Reviewer for the suggestion. We have added in the governing equations section (2.1), starting at line 62:

"In its 1D domain, FETCH3 allows for the vertical variation of the soil, root xylem, and stem xylem hydraulic parameters. As a result, the hydraulic conductivities, capacitances, root and stem xylem cross section areas, for example, are able to vary along the tree."

R2C10, L110: I found it confusing that "Sx" does not have the same units as "S" (also in equation 6). The same goes for "Aind" whose symbol suggests area units like "Ax", "As" and "Ar". How about explicitly writing "Ar/As" instead of "Aind"?

We agree with this comment, and have rewritten Equation (3), which now states "$S_x$" with the same units as the term "S" ($s^{-1}$). We have also modified the root lateral surface area index from "$A_{ind}$" to "$A_{lr}/A_x$ ", as suggested. We have modified these terms throughout the manuscript.

Equation 3:

$$C_x \frac{\partial \Phi_x}{\partial t} = \frac{d}{d\Phi_x} \left( \frac{\theta_x A_x}{A_s} \right) \frac{\partial \Phi_x}{\partial t} = \frac{\partial}{\partial z} \left[ K_x \frac{A_x}{A_s} \left( \frac{\partial \Phi_x}{\delta z} + \rho g \right) \right] - S_x \frac{A_x}{A_s}$$

Equation 5:

$$S_x \frac{A_x}{A_s} = T \, l(z)$$

R2C11, L122: The use of the subscript (z) for parameters like "r" but not variables like "theta" is a bit confusing. Why not using the subscript (z) for all variables and parameters that vary in space or mention it in a table? The same could be done for those varying in time.

Thank you for this point. We have adjusted Equation (4) and specified the dependence of the variables on space and time in the root water uptake formulation.

Equation 4 now reads:

$$S(z,t) = k_{s,rad} f\big(\theta_s(z,t)\big) . \frac{A_{ls}(z)}{A_s(z)} . \frac{r(z)}{\int_{z_{r_i}}^{z_{r_j}} r(z) \, dz} . (\Phi_s(z,t) - \Phi_r(z,t))$$

R2C12, L122: As it is presented, it is hard to understand why a vertical profile of relative root surface areas multiplies a normalized vertical profile of root mass in equation 4. Both seem to do the same job, don't they? In the discussion section, it would be interesting to discuss this formulation of the radial water flux in the broader context of existing models (e.g. De Jong Van Lier et al. (2008)).

The lateral root surface area index represents the lateral surface area of roots contained in a portion of reference soil area, whereas the root mass distribution represents the percentage of roots from the total root mass (not only accounting the for lateral roots) contained in the same soil reference area. The combination of these two terms in the root water uptake formulation (Eq. 4) scales the model to a more representative coupling of the roots with the soil. To clarify and better explain what this means, we added in section 2.2.1:

"$A_{ls}/A_s$ ($m^2_{root} \, m^{-2}_{ground}$) is an index defining the lateral root surface area per unit of ground, representing the root surface area taking up water from the soil. The vertical profile of root mass distribution represents the percentage of roots contained in different soil layer. The product of these

two terms provide the portion of roots contributing to the exchange of water between soil and roots; this changes with depth depending on how the roots are vertically distributed."

R2C13, L129: It is unclear if "T" is normalized by the soil, stem, or leaf surface area. Please clarify it when first introducing the variable.

We have added further clarification in the sentence, which now reads:

"(…) where T (m/s) is the transpiration rate defined per unit of ground area (…)"

R2C14, Figure 6: While I found the comparison with the analytical solution convincing, I am a bit more sceptical about the validation against transpiration rate data. To argue on the need for the new features of SPAC-3Hpy using a validation, I feel it would be relevant to show that the increased complexity and number of parameters is compensated by a substantial increase in the accuracy of the model predictions, possibly justified by the result of an Akaike test. Here the validation rather looks like an example run that fits reasonably well observed transpiration rates. However, with so many parameters involved, one could hardly imagine a poor fit of the transpiration data. This validation attempt that I feel is a bit oversold could be sent to appendices and replaced by a simple discussion contextualising the diversity of tree hydrodynamics models revolving around SPAC-3Hpy and explaining why we need this model, and which gap it will fill.

The case study is presented not as a validation test for FETCH3, but to show the ability of the model to reasonably reproduce experimental observations. The use of the dataset in Verma et al. (2014) is convenient because the observed sapflux rates were modelled with a similar set of equations as FETCH3, thereby not requiring a full calibration of the parameters. The case study shows that, with the same parameters, FETCH3 provides results similar to Verma et al. (2014), who used a finite difference scheme to solve the equations. The case study also serves to highlight the differences in the results that are obtained when including the storage capacity of the stem and a leaf area density distributing the transpiration along part of the stem.

For these reasons, that are better explained in the text, we prefer to keep the case study in the text. The additional section between the Introduction and the Model description will provide the context for the development of FETCH3, as suggested.

R2C15, Figure 6: It seems like there is a problem with inconsistent temporal scales in the top and bottom parts of figure 6. I realize that in a model with stem water capacitance, transpiration and water uptake may happen at slightly different times, but morning transpiration should come slightly before root water uptake, not the opposite. This issue is visible in the first peak. In the last peak, the temporal shift is in the opposite direction, possibly a bit off too. Please could you check the time scales?

We have fixed Figure 6, and it now presents the correct horizontal axis. The figure got shifted when being converted to a higher resolution.

Best of luck with the next steps.

Valentin

**References:**

**Couvreur V, Ledder G, Manzoni S, Way DA, Muller EB, Russo SE** (2018) Water transport through tall trees: A vertically explicit, analytical model of xylem hydraulic conductance in stems. Plant, Cell & Environment **41:** 1821-1839

**De Jong Van Lier Q, Van Dam JC, Metselaar K, De Jong R, Duijnisveld WHM** (2008) Macroscopic root water uptake distribution using a matric flux potential approach. Vadose Zone J. **7:** 1065-1078

**Kennedy D, Swenson S, Oleson KW, Lawrence DM, Fisher R, Lola da Costa AC, Gentine P** (2019) Implementing Plant Hydraulics in the Community Land Model, Version 5. Journal of Advances in Modeling Earth Systems **11:** 485-513

**Meunier F, Rothfuss Y, Bariac T, Biron P, Richard P, Durand J-L, Couvreur V, Vanderborght J, Javaux M** (2017) Measuring and Modeling Hydraulic Lift of Lolium multiflorum Using Stable Water Isotopes. Vadose Zone J.**:** 15 pp.

**Sperling O, Shapira O, Cohen S, Tripler E, Schwartz A, Lazarovitch N** (2012) Estimating sap flux densities in date palm trees using the heat dissipation method and weighing lysimeters. Tree Physiol. **32:** 1171-1178

---

## Author Response (AR1)

**Reviewer 1**

Reviewers' comments are in black font, our responses are in blue, and green is for new and revised text; to assist with navigation, we use codes, such as R1C1 for Reviewer 1 Comment 1.

The study describes the development of a new SPAC model. The study is of great relevance to hydrology ecohydrology and ecosystem modelling and clearly within the scope of the journal. The model couples the partial differential equations that describe water flow in the soils, and plants.

R1C1: My main concern regarding the paper is its novelty. The authors should more clearly illustrate the new features of the present model that have not been previously reported. For example, the plant water transport is very similar to FETCH2, the whole model set-up very similar to Huang, C-W et al., 2017 New phytologist (already cited in the paper). The authors should more clearly present the main novelties of the present model. I am not saying that the model is not novel, but rather the novelty needs to be better described in the manuscript.

The novelty of the model was one of the main concerns raised by both reviewers and the editor, especially in relation to the differences from FETCH and FETCH2. We decided to change the name of the model to FETCH3, and this is used in this document. The former acronym from FETCH no longer reflects the nature of the model and numerical scheme used to solve the equations, so we have changed it to "Finite-difference Ecosystem-scale Tree Crown Hydrodynamics", which easily relates to the developments that FETCH has had over the years. In addition, we have added further clarifications on how the three versions differentiate from each other.

The differences between FETCH2 and the present model, FETCH3, are summarized in section 2.1, starting at line 77, as:

"The Finite-difference Ecosystem-scale Tree Crown Hydrodynamics (FETCH3) builds upon FETCH2 (Mirfenderesgi et al., 2016, 2018), which is based on its precursor, the Finite Element Tree Crown Hydrodynamics (FETCH) model (Bohrer et al., 2005). FETCH simulates water flow along a tree's stem and branches accounting for the branch structure in three dimensions. Simulating the three-dimensional tree crown structure is computational demanding and can solely be applied to a single tree. As a result, FETCH2 was developed to offer a more mechanistic approach that could be scaled to entire ecosystems. To achieve this, FETCH2 simplifies branches along the vertical direction, leading to a 1D model; the equations in FETCH2 are solved using a finite difference scheme (Mirfenderesgi et al., 2016).

Similarly to FETCH and FETCH2, FETCH3 assumes that the water movement in the xylem resembles flow in porous media; as in FETCH2, a macroscopic approach is used to simulate the water fluxes across the soil, roots, and stems with the fluxes being described in one dimension along the vertical direction (Fig. 1). As a development from FETCH2, FETCH3 presents a clearer link between the three different components of the system (i.e., soil, roots and stem), based on the conservation of water in each of the components, as derived in the Supplementary Material. As a result, when combined, the quantities in the equations for the roots and stem are scaled to a reference ground area, consistently with the Richardson-Richards equation for the soil. This guarantees the conservation of mass as water flows from one component to the other. The system of equations in FETCH3 is also solved differently from FETCH2. As described in detail in the Supplementary Material, the equations in FETCH3 are discretised using the method by Celia et al. (1990) generating a system of algebraic equations combined into a single matrix, that is solved at the same time to guarantee the conservation of mass across the whole system comprising soil, roots and stem."

The water transport described in Huang et al. (2017) is different from the one in FETCH3. Huang et al. (2017) simplified their system by excluding a detailed description of the roots, assuming that the root water potential reaches very rapidly hydrostatic conditions dependent on the water potential at the base of the stem.

We have added the differences between the modelling approaches (i.e., electric circuit equivalence, porous media) in the largely revised introduction.

R1C2: Regarding the model implementation itself, it is great to see comparisons between analytical solutions and previous numerical solutions for model confirmation.

We thank the reviewer for the encouraging comment.

R1C3: Regarding the complexity of the model, my main comment is that the model formulation seems incomplete for a SPAC model, as it mostly neglects the atmospheric component. I would expect a SPAC model to be forced with meteorological variables. At the current state tree transpiration is provided as a boundary condition, instead of being computed prognostically. The authors can consider expanding the model to have this capability.

FETCH3 is designed to allow the implementation of different transpiration formulations in Equation (5). For our study, we used a simpler transpiration function for section 3.1, where we tested the numerical scheme against analytical solutions, and used the Penman-Monteith equation (Allen et al., 1998) combined with a stomatal conductance formulation (Jarvis, 1976) for the test case with experimental data. Both Penman-Monteith and stomatal conductance formulations are forced by meteorological variables; the stomatal conductance also depends on the stem water potential, and thus transpiration is not provided as a boundary condition but is calculated by the model.

The equations can be found in the Supplementary Material, section S.3. To clarify these aspects, we added in section 2.2.1 starting at line 165:

"The water lost to the atmosphere is calculated using a transpiration function that depends on meteorological variables and limits the amount of water leaving the stomata as a function of the stem water potential. FETCH3 allows for the implementation of different transpiration functions, and a complete description of the transpiration formulation applied in this study is in the Supplementary Material, section S.3."

R1C4: To my understanding, the model in its current form, can only simulate a single dry-down period as no infiltration is implemented. This is something that the authors might want to include in the model as it cannot be currently used for continuous long term simulations.

FETCH3 includes infiltration implemented as a boundary condition at the top of the soil column, as described in the Supplementary Material, section S.2.2 (page 12). The term qinf was modelled according to the rainfall rate and soil moisture at the surface layer (section 3.2.2). To further clarify this capability, we added more details in sections 2.1 and 3.2.2:

(Section 2.1, starting at line 96)

"The water flow in the soil is modelled using the Richardson-Richards equation with a term simulating the exchange of water between the soil and the roots. This term is a function of the difference in water potential between the soil and root layers; it thus results in a water sink during the day, when the water potential in the roots is low due to water loss by transpiration, but may act

as a source of water to the soil during some nights, depending on the water content in different soil layers. The boundary conditions at the top and bottom of the soil column can be expressed as a flux or a value of soil water potential (refer to the Supplementary Material, section S.2.2)."

(Section 3.2.2, starting at line 294)

"At the surface, measured rainfall was used as a flux boundary condition to compute soil water infiltration (refer to the Supplementary Material, section S.2.2)."

R1C5: A finite difference method was used to solve the Richards' equation for both soils and plants. This numerical formulation does not guarantee mass conservation. As a sanity check I would advise the authors to report the total water mass conservation. Given the accuracy of the model in recovering the analytical solution, I am confident that any discrepancy is negligible but worth reporting nevertheless.

We thank the reviewer for the suggestion. We have added the water mass balance for the unsteady and steady cases, and for the case study.

For the unsteady and steady cases, only the stem section of the tree was considered. For the two unsteady cases that we considered, the mass balance error was calculated as the total water that entered at the bottom of the tree minus the total transpiration and the change in water storage during the simulation. We expressed the mass balance error as a percentage of the total water that entered the bottom of the tree. For the two unsteady cases, the water mass error was 0.05% of the total mass of water that entered the tree, for a simulation period of 2 days.

For the steady case, the flux entering at the bottom of the tree and the transpiration rate are expected to be equal, with no changes in storage. We checked that in the numerical calculations, at steady state, the flux of water entering at the bottom of the tree and the transpiration rates were close to each other, and changes in storage were approximately null. At steady state, transpiration was equal to 99.97% of the total flux entering the tree, and the difference in storage between time steps was equal to  $-2.77 \cdot 10^{-18} \text{ m}^3$ .

For the case study, we expressed the mass balance error in the tree as the sum of storages in the tree (in the root and stem xylem) plus the root water uptake minus transpiration. We expressed this error as a percentage of the total amount of water that infiltrated the soil during the simulation. The water mass balance error in the tree was equal to 0.16% of the infiltration over a period of approximately 5 months (1st January to 4th June). In the soil, the mass balance error was calculated as the total soil water storage plus the sum of total water flux at the bottom of the soil (since the boundary condition at the soil bottom is a constant water potential) and infiltration, minus the root water uptake. This difference was equal to 0.30% of the total infiltration over the simulation period.

We included the following additions in the manuscript. For the unsteady state solution with the sink term depending only on time (starting at line 236):

"The error followed the pattern of transpiration, reaching its peak during day time and corresponding to a maximum error of 0.09% of the exact solution. The mass balance error equalled 0.05% of the total water entering the tree during the simulated 2 days."

For the unsteady state solution, with a sink term depending on both time and vertical coordinate (z) (starting at line 238):

"The error for this case is higher than for the previous case, with a maximum value that is about 0.2% of the exact solution, with a mass balance error equal to 0.05% of the total water entering the tree during the simulated 2 days."

For the steady state solution (starting at line 245):

"For a 6-meter-high tree, the error of the numerical solution increases with elevation reaching approximately  $0.4 \cdot 10^{-3}$  MPa at the tree top, being 0.4% of the exact value. According to the steady-state condition, the differences in storage between the last two consecutive model time steps approached zero and were equal to -2.77e-18 m3, with transpiration equalling 99.97% of the total flux entering the tree."

For the case study (starting at line 309):

"The model predictions for sap-flux during the day compared well with the observations during the entire measurement period (Fig.5a), reaching a R2 value of 0.74. The total mass balance error in the soil represented -0.30% of total infiltration, and it was calculated as the change in soil water storage minus the difference between the flux entering (bottom boundary condition and infiltration) and exiting the soil (root water uptake). In the tree (root and stem xylem), the water mass error was - 0.16% of the total infiltration, and was calculated as the change in water storage (in the stem and root xylem) minus the difference between the fluxes entering (root water uptake) and exiting (transpiration) the tree."

R1C6: A discussion point that might need to be better addressed is the added benefit of the vertically distributed, computationally expensive solution. Many ecosystem models lump tree hydraulics with a small number of resistances (commonly soil to root, root to leaf and leaf to atmosphere) or a combination of a small number of resistors and capacitors (e.g., ED2 model – Trugman, Anna T., et al. "Leveraging plant hydraulics to yield predictive and dynamic plant leaf allocation in vegetation models with climate change." Global change biology12 (2019): 4008-4021.). This approach is definitely more computationally parsimonious, and less data demanding as all plant hydraulic traits are lumped. A discussion of pros/cons would benefit the paper.

We have further discussed in the revised introduction the differences between the modelling approaches and addressed the potential benefits that may offset the more computational heavy approach of a vertically detailed model (such as FETCH3), compared to a simple single layer (or multiple single layers, as is Trugman's) more computationally cheap modelling approach. This will clarify the pros and cons of each approach. We also added a few sentences in the Conclusion to highlight some of the benefits of FETCH3.

R1C7: In page 3 lines 64 and 73, I would advise the authors to rephrase the term "lumped" as it might lead to confusion as the model is at least in 1D distributed.

We thank the reviewer for the suggestion. We re-phrased these parts:

(Section 2.1, starting at line 84):

"Similarly to FETCH and FETCH2, FETCH3 assumes that the water movement in the xylem resembles flow in porous media; as in FETCH2, a macroscopic approach is used to simulate the water fluxes across the soil, roots, and stems with the fluxes being described in one dimension along the vertical direction (Fig. 1)."

(Section 2.1, starting at line 108):

"The 3D root architecture is **scaled** along the vertical dimension using a vertical mass distribution of the roots and an index that summarizes the extent of lateral root area per unit of ground area (Quijano and Kumar, 2015)."

R1C8: Looking at the Python code, I noticed that object orientation was hardly ever used, that would be great for a modular model design that can be used to "plug-in" additional modules in the future (e.g., radiative transfer schemes, photosynthesis, phloem transport etc). The authors might consider in the future reconstruction of the code.

We thank the reviewer for this suggestion. This is certainly something to be further developed and implemented. At this stage, our focus was to provide an open access code with a verified and tested solution of the system of Richardson-Richards equations. We have now provided an updated version of the model with a more modular structure and with a vertically detailed transpiration function in an easy to replace function. This highlights the importance of including multiple vertical layers, with a structure that allows for interactions between radiation and atmospheric conditions within the canopy and different storage and stomata restrictions at different vertical levels.

**Reviewer 2**

Reviewers' comments are in black font, our responses are in blue and green is for new and revised text; to assist with navigation, we use codes, such as R2C1 for Reviewer 2 Comment 1.

**General comments:**

In this manuscript, the authors describe the one-dimensional numerical model of tree hydrodynamics SPAC-3Hpy with explicit Richards-like solutions of flow in soil, root, and stem domains. It is proposed to validate it against analytical solutions of water flow in porous media, and against tree transpiration data. The scope is of great interest to the readership of Geoscientific Model Development. Overall, the study is interesting, well explained and provides clear illustrations.

**We thank Dr Couvreur for his overall positive evaluation of our work.**

My main concerns regarding publication in GMD are (i) the need for clarifications and small improvements that would facilitate the interpretation of the manuscript, (ii) the validation against transpiration data that I think does not meet the quality of the rest of the manuscript, and (iii) the lack of a proper contextualization of the model relative to the galaxy of existing tree hydrodynamics models that could be developed in a discussion section.

We have addressed the 3 points raised by the Reviewer in the answers to the specific comments below. In relation to the discussion section, we have included in the revised introduction clarification between the differences in modelling approaches. This will clarify the pros and cons of each approach to provide the context requested by both Reviewers.

To provide a clearer contextualization of the model, we decided to change the name of the model to FETCH3, and this is used in this document. The former acronym from FETCH no longer reflects the nature of the model and numerical scheme used to solve the equations, so we have changed it to "Finite-difference Ecosystem-scale Tree Crown Hydrodynamics", which easily relates to the developments that FETCH has had over the years.

**Specific comments:**

R2C1 (L11): The term "hydroactive" xylem is a bit odd as the process of water flow in xylem is passive as described in the introduction of the paper. Also, throughout the paper the authors seem to use the term "xylem" when referring to the stem (e.g. also at L84). This is confusing as there is xylem in roots and leaves too. I would suggest to revise the wording.

Thank you for the suggestion. We removed the term "hydroactive" xylem and introduced the terms "stem xylem" and "root xylem" to improve the reading.

R2C2, L12: I am enthused that the authors made their model open access. It also seems that the code is open source, which is even better I believe. If that is correct, why not making it explicit in the text?

Thank you for your comment. We added in the Introduction, starting at line 66:

"The aim of this study is to present FETCH3, an open source and open access tree hydrodynamic model for the simulation of the temporal and vertical dynamics of water storage and fluxes from the soil to the atmosphere, accounting for the vegetation response to environmental conditions and soil water availability."

R2C3, L21-22: I think the sentence needs to be completed. Upward flow of water in 20 m trees would also be possible without the tension-cohesion mechanism, using an osmotic pressure difference of 0.2 MPa between soil and xylem. The tension-cohesion theory explains that this upward flow is possible without an osmotic driving force, along a continuum of liquid water under tension.

We added in the Introduction (starting at line 20): "Most models that explicitly resolve the movement of water within the plant system rely on the cohesion-tension theory, which explains how water can be transferred upward from the soil to the atmosphere across a tree height of several meters, in the absence of osmotic pressure differences (Couvreur et al., 2018)."

R2C4, L40-42: Root water uptake compensation and hydraulic redistribution can also be modelled with "electric circuit" models that do not account for the plant water capacitance (see e.g. Meunier et al. (2017); Kennedy et al. (2019)). A major progress with the capacitance is that water fluxes along the stem do not have to be vertically equal simultaneously (e.g. as observed in Sperling et al. (2012) in palm) and that the integral of root water uptake does not have to match the integral of transpiration at a given time due to variations in stem water storage.

Thank you for your comment. We have included in the revised introduction clarification between the differences in modelling approaches. This will clarify the pros and cons of each approach to provide the context requested by both Reviewers.

R2C5, L63-64: Similarities with FETCH and FETCH2 are mentioned but differences are unclear and should be clearly explained. A discussion section could as well discuss the specific interest of SPAC-H3py relative to the broader diversity of tree hydrodynamics models.

The differences between the present model, FETCH3, and FETCH2 are summarized in section 2.1, starting at line 77, as:

"The Finite-difference Ecosystem-scale Tree Crown Hydrodynamics (FETCH3) builds upon FETCH2 (Mirfenderesgi et al., 2016, 2018), which is based on its precursor, the Finite Element Tree Crown Hydrodynamics (FETCH) model (Bohrer et al., 2005). FETCH simulates water flow along a tree's stem and branches accounting for the branch structure in three dimensions. Simulating the three-dimensional tree crown structure is computational demanding and can solely be applied to a single tree. As a result, FETCH2 was developed to offer a more mechanistic approach that could be scaled to entire ecosystems. To achieve this, FETCH2 simplifies branches along the vertical direction, leading to a 1D model; the equations in FETCH2 are solved using a finite difference scheme (Mirfenderesgi et al., 2016).

Similarly to FETCH and FETCH2, FETCH3 assumes that the water movement in the xylem resembles flow in porous media; as in FETCH2, a macroscopic approach is used to simulate the water fluxes across the soil, roots, and stems with the fluxes being described in one dimension along the vertical direction (Fig. 1). As a development from FETCH2, FETCH3 presents a clearer link between the three different components of the system (i.e., soil, roots and stem), based on the conservation of water in each of the components, as derived in the Supplementary Material. As a result, when combined, the quantities in the equations for the roots and stem are scaled to a reference ground area, consistently with the Richardson-Richards equation for the soil. This guarantees the conservation of mass as water flows from one component to the other. The system of equations in FETCH3 is also solved differently from FETCH2. As described in detail in the Supplementary Material, the equations in FETCH3 are discretised using the method by Celia et al. (1990) generating a system of algebraic

equations combined into a single matrix, that is solved at the same time to guarantee the conservation of mass across the whole system comprising soil, roots and stem."

Additionally, we have included in the revised introduction clarification between the differences in modelling approaches. This will clarify the pros and cons of each approach.

R2C6, Figure 1: It is good that variables appear in figure 1, so that connections to the equations can be done. However, several of these variables have not been defined at this point, which complicates the interpretation of the figure. There is room for it in the caption, or in the main text already.

**The caption of Figure 1 has been modified as:**

"Representation of the coupling process between soil, root xylem, and stem xylem applied in the model, where As represents a reference ground area, dz an infinitesimal depth over an area (m), z the vertical coordinate (m), V volume of soil (m3),  $\rho$  the density of water (kg m-3), Fin (kg s-1) the water fluxes entering and Fout (kg s-1) exiting the volume, Ar/As (m2root m-2ground) the root xylem cross area index, Ax/As (m2xylem m-2ground) the stem xylem cross area index, S (s-1) the rate at which water is extracted from the soil and enter the root xylem, and Sx (m2 s-1) the flow of water leaving the stem per unit of length due to transpiration."

R2C7, L70-72: At this point it is not clear if radial resistances between the bulk soil, root surface and root xylem are considered. It is particularly important to specify it in the overview as they are frequently viewed as the largest resistances in series of the soil plant hydraulic continuum.

FETCH3 considers the radial resistances (which includes a soil moisture stress function), lateral surface area of the roots and root mass distribution when computing the root water uptake. We have added starting at line 103 of the overview section (2.1):

"Water fluxes within roots are likewise modelled with a Richardson-Richards type equation with the same term (of the opposite sign) representing water exchange between roots and soil. Soil and roots are coupled through this term, such that a sink of water in the soil is a source of water in the roots, and vice versa. The transfer of water between the soil and the roots is modulated by a conductance, representing the radial resistance between the bulk soil, roots surface, and root xylem, and a stress function, accounting for the reduction of the root water uptake associated with different soil moisture conditions possibly leading to water and oxygen stress."

R2C8, L99: I realize it is implicit, but probably worth specifying that the "soil capacitance" is the "soil water capacitance". The same goes for the root and stem water capacitances.

**We have changed the terms throughout the manuscript as suggested.**

R2C9, L99: Going through the equations I frequently wondered if parameters were constants or could vary in space. For instance, stem cross-sectional area, saturated hydraulic conductivity (here Km I believe) and sensitivity to cavitation (that I guess relate to the water capacitance parameter) vary along tree stems with substantial consequences on the nonlinear vertical water potential profile (Couvreur et al., 2018). It would be worth discussing briefly if such vertical variations of hydraulic parameters can be accounted for in SPAC-3Hpy.

We thank the Reviewer for the suggestion. We have added in the governing equations section (2.1), starting at line 88:

"In its 1D domain, FETCH3 allows for the vertical variation of the soil, root xylem, and stem xylem hydraulic parameters, which are able to vary along the tree."

R2C10, L110: I found it confusing that "Sx" does not have the same units as "S" (also in equation 6). The same goes for "Aind" whose symbol suggests area units like "Ax", "As" and "Ar". How about explicitly writing "Ar/As" instead of "Aind"?

The term " $S_x$ " is different from "S". The sink term in the soil, S, refers to the amount of water that is removed from a volume of soil. Conversely,  $S_x$  does not depend of a volume of soil, because it is related to the water balance of a portion of xylem with infinitesimal height dz;  $S_x$  also depends on the leaf area, and not directly to the cross-sectional area of the stem xylem, given that water is transpired through the leaves. Thus, " $S_x/A_s$ " in Equation (3) is proportional to the area of the leaves and can be expressed as  $S_x$ = I(z)  $A_s$  T (Equation 5), with I(z) in m2 m-1 and T, expressed as water transpired per unit of ground, in m s-1. Therefore, we decided to keep the term " $S_x$ " as it is.

We have modified the root lateral surface area index from " $A_{ind}$ " to " $A_{lr}/A_x$ ", as suggested. We have modified this term throughout the manuscript.

R2C11, L122: The use of the subscript (z) for parameters like "r" but not variables like "theta" is a bit confusing. Why not using the subscript (z) for all variables and parameters that vary in space or mention it in a table? The same could be done for those varying in time.

Thank you for this point. We have adjusted Equation (4) and specified the dependence of the variables on space and time in the root water uptake formulation.

Equation 4 now reads:

$$S(z,t) = k_{s,rad} f(\theta_s(z,t)) \cdot \frac{A_{ls}(z)}{A_s(z)} \cdot \frac{r(z)}{\int_{z_{r_i}}^{z_{r_j}} r(z) \, dz} \cdot (\Phi_s(z,t) - \Phi_r(z,t))$$

R2C12, L122: As it is presented, it is hard to understand why a vertical profile of relative root surface areas multiplies a normalized vertical profile of root mass in equation 4. Both seem to do the same job, don't they? In the discussion section, it would be interesting to discuss this formulation of the radial water flux in the broader context of existing models (e.g. De Jong Van Lier et al. (2008)).

The lateral root surface area index represents the lateral surface area of roots contained in a portion of reference soil area, whereas the root mass distribution represents the percentage of roots from the total root mass (not only accounting the for lateral roots) contained in the same soil reference area. The combination of these two terms in the root water uptake formulation (Eq. 4) scales the model to a more representative coupling of the roots with the soil. To clarify and better explain what this means, we added in section 2.2.1 starting at line 161:

"AIs/As (m2root m-2ground) is an index defining the lateral root surface area per unit of ground, representing the root surface area taking up water from the soil. The vertical profile of root mass distribution represents the percentage of roots contained in different soil layer. The product of these two terms provide the portion of roots contributing to the exchange of water between soil and roots; this changes with depth depending on how the roots are vertically distributed."

R2C13, L129: It is unclear if "T" is normalized by the soil, stem, or leaf surface area. Please clarify it when first introducing the variable.

We have added further clarification in the sentence, which now reads (section 2.2.1, line 170):

**"(...) where T (m/s) is the transpiration rate defined per unit of ground area (...)"**

R2C14, Figure 6: While I found the comparison with the analytical solution convincing, I am a bit more sceptical about the validation against transpiration rate data. To argue on the need for the new features of SPAC-3Hpy using a validation, I feel it would be relevant to show that the increased complexity and number of parameters is compensated by a substantial increase in the accuracy of the model predictions, possibly justified by the result of an Akaike test. Here the validation rather looks like an example run that fits reasonably well observed transpiration rates. However, with so many parameters involved, one could hardly imagine a poor fit of the transpiration data. This validation attempt that I feel is a bit oversold could be sent to appendices and replaced by a simple discussion contextualising the diversity of tree hydrodynamics models revolving around SPAC-3Hpy and explaining why we need this model, and which gap it will fill.

The case study is presented not as a validation test for FETCH3, but to show the ability of the model to reasonably reproduce experimental observations. The use of the dataset in Verma et al. (2014) is convenient because the observed sapflux rates were modelled with a similar set of equations as FETCH3, thereby not requiring a full calibration of the parameters. The case study shows that, with the same parameters, FETCH3 provides results similar to Verma et al. (2014), who used a finite difference scheme to solve the equations. The case study also serves to highlight the differences in the results that are obtained when including the storage capacity of the stem and a leaf area density distributing the transpiration along part of the stem.

For these reasons, that we tried to better explained in the text, we prefer to keep the case study as part of the main text.

**The additions in the largely revised Introduction will provide the context for the development of FETCH3, as suggested.**

R2C15, Figure 6: It seems like there is a problem with inconsistent temporal scales in the top and bottom parts of figure 6. I realize that in a model with stem water capacitance, transpiration and water uptake may happen at slightly different times, but morning transpiration should come slightly before root water uptake, not the opposite. This issue is visible in the first peak. In the last peak, the temporal shift is in the opposite direction, possibly a bit off too. Please could you check the time scales?

We have fixed Figure 6, and it now presents the correct horizontal axis. The figure got shifted when being converted to a higher resolution.

Best of luck with the next steps.

Valentin

**References:**

**Couvreur V, Ledder G, Manzoni S, Way DA, Muller EB, Russo SE** (2018) Water transport through tall trees: A vertically explicit, analytical model of xylem hydraulic conductance in stems. Plant, Cell & Environment **41**: 1821-1839

**De Jong Van Lier Q, Van Dam JC, Metselaar K, De Jong R, Duijnisveld WHM** (2008) Macroscopic root water uptake distribution using a matric flux potential approach. Vadose Zone J. **7:** 1065-1078

Kennedy D, Swenson S, Oleson KW, Lawrence DM, Fisher R, Lola da Costa AC, Gentine P (2019) Implementing Plant Hydraulics in the Community Land Model, Version 5. Journal of Advances in Modeling Earth Systems **11**: 485-513

Meunier F, Rothfuss Y, Bariac T, Biron P, Richard P, Durand J-L, Couvreur V, Vanderborght J, Javaux M (2017) Measuring and Modeling Hydraulic Lift of Lolium multiflorum Using Stable Water Isotopes. Vadose Zone J.: 15 pp.

**Sperling O, Shapira O, Cohen S, Tripler E, Schwartz A, Lazarovitch N** (2012) Estimating sap flux densities in date palm trees using the heat dissipation method and weighing lysimeters. Tree Physiol. **32:** 1171-1178